# Cell type-dependent differential activation of ERK by oncogenic KRAS in colon cancer and intestinal epithelium

Raphael Brandt [1,9], Thomas Sell [1,2,9], Mareen Lüthen [1,3], Florian Uhlitz[1,2,3], Bertram Klinger[1,2], Pamela Riemer [1], Claudia Giesecke-Thiel [4,5], Silvia Schulze[1], Ismail Amr El-Shimy [1,2], Desiree Kunkel[6], Beatrix Fauler[5], Thorsten Mielke [5], Norbert Mages[5], Bernhard G. Herrmann [5,7], Christine Sers [1,3,8], Nils Blüthgen [1,2,3,8,10] & Markus Morkel [1,3,8,10]

Oncogenic mutations in KRAS or BRAF are frequent in colorectal cancer and activate the ERK kinase. Here, we find graded ERK phosphorylation correlating with cell differentiation in patient-derived colorectal cancer organoids with and without KRAS mutations. Using reporters, single cell transcriptomics and mass cytometry, we observe cell type-specific phosphorylation of ERK in response to transgenic KRAS$^{G12V}$ in mouse intestinal organoids, while transgenic BRAF$^{V600E}$ activates ERK in all cells. Quantitative network modelling from perturbation data reveals that activation of ERK is shaped by cell type-specific MEK to ERK feed forward and negative feedback signalling. We identify dual-specificity phosphatases as candidate modulators of ERK in the intestine. Furthermore, we find that oncogenic KRAS, together with β-Catenin, favours expansion of crypt cells with high ERK activity. Our experiments highlight key differences between oncogenic BRAF and KRAS in colorectal cancer and find unexpected heterogeneity in a signalling pathway with fundamental relevance for cancer therapy.

[1] Institute of Pathology, Charité Universitätsmedizin Berlin, Charitéplatz 1, 10117 Berlin, Germany. [2] IRI Life Sciences, Humboldt University Berlin, Philippstrasse 13, 10115 Berlin, Germany. [3] German Cancer Consortium (DKTK), German Cancer Research Center (DKFZ), 69120 Heidelberg, Germany. [4] Department of Cell Biology, German Rheumatism Research Center, Leibniz Institute, Berlin, Germany. [5] Max Planck Institute for Molecular Genetics, Ihnestr. 73, 14195 Berlin, Germany. [6] Berlin-Brandenburg Center for Regenerative Therapies (BCRT), Charité - Universitätsmedizin Berlin, Campus Virchow-Klinikum, Augustenburger Platz 1, 13353 Berlin, Germany. [7] Institute for Medical Genetics, Charité Universitätsmedizin Berlin, Hindenburgdamm 30, 12203 Berlin, Germany. [8] Berlin Institute of Health (BIH), Anna-Louise-Karsch-Str. 2, 10178 Berlin, Germany. [9]These authors contributed equally: Raphael Brandt, Thomas Sell. [10]These authors jointly supervised this work: Nils Blüthgen, Markus Morkel. Correspondence and requests for materials should be addressed to N.B. (email: nils.bluethgen@charite.de) or to M.M. (email: markus.morkel@charite.de)

Multiple signalling pathways, including the mitogen-activated protein kinase (MAPK) and Wnt/β-catenin cascades, form a network controlling cellular turnover in the intestinal epithelium[1]. Collectively, activities within the signalling network control stem cell maintenance, cell proliferation, differentiation into absorptive enterocyte and secretory cells, and apoptosis. Wnt/β-catenin and MAPK activities are regionalised within the folded single-layered intestinal epithelium. Both are high in crypts harbouring stem cells and low in differentiated cells that have migrated away from the crypt base. Oncogenic mutations activating β-catenin and MAPK perturb intestinal homeostasis and thereby drive colorectal cancer (CRC) initiation and progression.

MAPK modules transduce signals downstream of receptor tyrosine kinases and RAS family GTPases. The consecutive RAF, MEK and ERK kinases represent a MAPK module frequently activated in cancer. ERK can phosphorylate and activate a series of transcription factors orchestrating a complex cellular response that often is pro-proliferative[2]. In the normal intestine, EGFR to ERK signalling is initiated by ligands from the crypt micro-environment, which are secreted by e.g. epithelial Paneth cells of the small intestine, Reg4+ secretory niche cells of the large intestine, or adjacent fibroblasts[3,4]. In CRC, ERK activity is supposedly more cell-autonomous due to oncogenic mutations activating KRAS, NRAS or BRAF (found in 45, 5 and 10% of CRCs, respectively)[5,6], or by de novo expression of EGFR ligands such as amphiregulin[7]. Signal transduction to ERK is a main determinant of cancer development and therapy response[5,6,8].

Recent studies suggest that the relationship between ERK-activating mutations, ERK activity and phenotypic outcome in CRC is complex. Firstly, mutations in KRAS and BRAF are associated with distinct CRC development routes: KRAS, but not BRAF, mutations frequently occur as secondary events after mutations activating Wnt/β-catenin in the conventional CRC progression sequence[9,10]. Conversely, BRAF, and less frequently KRAS, mutations precede activation of Wnt/β-catenin in the alternative serrated progression route[11,12]. The observed disequilibrium between KRAS and BRAF mutations in the conventional vs. serrated pathways of CRC evolution suggest the existence of functional differences, resulting in distinctive effects on clinical course and treatment efficacy[13]. Secondly, ERK activity appears to be heterogeneous in genetically identical CRC cells. Cells at the invasive front frequently exhibited higher ERK phosphorylation levels compared with cells in central areas of the same cancer, and CRCs with activating KRAS mutations also showed heterogeneous ERK activity[14]. Previous studies already showed heterogeneous Wnt/β-catenin activity in cancer specimens, suggesting a more general paradigm of graded pathway activities in CRC[15]. Furthermore, CRC cells have been shown to exhibit functional differences within a cancer, as only few CRC cells, so-called cancer stem cells, could initiate new tumours in xenografts[16–19]. Gradients of surface markers, such as EphB2, were found to distinguish CRC stem cells with high malignant potential[20]. CRC subtypes can share similarities with cell types of the normal crypt, such as stem cells, enterocytes or secretory cells in bulk cell analysis[21]. Finally, because of variable signal transduction and differentiation states, genetically identical CRC clones exhibit variable proliferative potential and therapeutic response[22].

Experimental techniques with cellular resolution, ranging from fluorescent reporters[23] to single-cell transcriptome analyses[24,25] and mass cytometry[26] hold the promise to disentangle the relationship between oncogenes, cell differentiation states and cell-signal transduction while taking into account cellular heterogeneity. Here, we ask whether oncogenic forms of KRAS or BRAF show cell-to-cell heterogeneity in their proclivity to activate ERK.

For this, we use patient-derived and mouse transgenic organoid cultures that maintain the cell hierarchy of tissue in vitro[27]. We assess signalling network states with cellular resolution by mass cytometry and use BRAF[V600E] and KRAS[G12V] transgenes to assess immediate impact of the oncogenes on cell-signal transduction, gene expression programmes and phenotypic outcome. We discover strong functional differences between the BRAF and KRAS oncogenes and find that signal transduction from KRAS to ERK is shaped by different strengths of feed forward and negative feedback in a cell type-specific manner.

## Results

**Graded ERK activity in KRAS-mutant CRC organoids**. To investigate whether oncogenic KRAS enforces constitutive activity of MEK and ERK kinases, we examined patient-derived three-dimensional CRC organoid cultures by immunohistochemistry. We found heterogeneous phosphorylation of both, MEK and ERK, in organoids with no mutations in the EGFR-RAS-ERK cascade (line OT326), as well as in KRAS-mutant organoids (line OT227, carrying a KRAS[G13D] mutation) (Fig. 1a).

We next used mass cytometry to analyse cell differentiation markers and MEK and ERK phosphorylation side-by side in the patient-derived organoids (Fig. 1b). We selected the two organoid lines used above, as well as line OT302, harbouring a KRAS[G12D] mutation. We found that cells of all three organoid lines formed gradients with respect to levels of EphB2, a known marker of CRC hierarchies linked to metastasis and therapy response[20]. In two of the three lines (OT326 and OT302), a substantial proportion of EphB2-low cells was marked by cleaved Caspase 3, suggesting apoptotic removal of cells at the end of their life span. Intriguingly, all three lines displayed gradients of phosphorylated MEK and ERK that were largely congruent with EphB2. These results indicate that patient-derived CRC organoids contain phosphorylation gradients of MEK and ERK kinases along an axis defined by cell differentiation. The observed gradient formed regardless of oncogenic activation of the upstream KRAS GTPase, and in the absence of tumour stroma that is not present in the organoids.

**BRAF[V600E], but not KRAS[G12V], induces strong ERK activity**. As MEK and ERK activities were graded along a differentiation axis in patient-derived CRC organoids irrespective of mutational status of KRAS, we asked whether oncoproteins activating the MEK-ERK signalling axis exert their activities in a cell type-specific manner. To study this question, we employed intestinal organoids of transgenic mice carrying doxycycline-inducible single copy constructs encoding tdTomato linked to KRAS[G12V], BRAF[V600E], or firefly luciferase (FLUC) as a control in the Gt (ROSA26)Sor locus (Fig. 2a)[28,29].

We initiated organoid cultures by embedding intestinal crypts from FLUC-, KRAS[G12V]-, and BRAF[V600E]-inducible mice into extracellular matrix, as described before[27]. When we induced oncoprotein production by adding doxycycline to the culture media, BRAF[V600E] led to irreversible disintegration of organoids within 1–2 days, whereas transgenic KRAS[G12V] or the FLUC control protein were well tolerated, even after several passages (Fig. 2b). To examine whether the BRAF oncogene has detrimental effects on the epithelium beyond the previously reported loss of stem cells[29,30], we examined histology of the induced organoids at ultrastructural level using transmission electron microscopy (Fig. 2c). We found that control and KRAS[G12V]-induced organoids showed the expected tissue structure, that is, a single-layered polarised epithelium with continuous apical and basal surfaces as well as a brush border at the apical side. Desmosomes, providing lateral cell adhesion, were

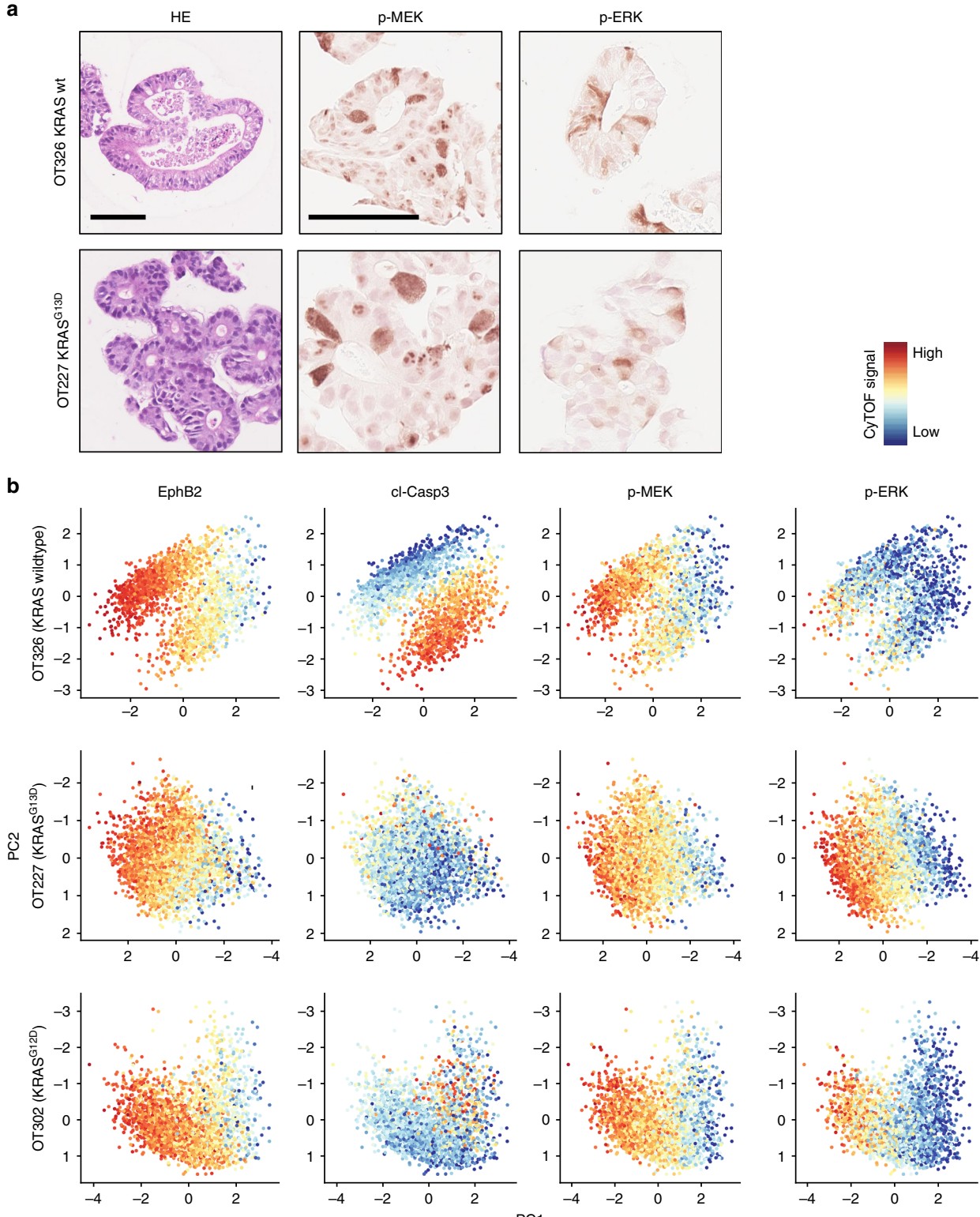

**Fig. 1** Graded MEK and ERK phosphorylation in patient-derived organoids. **a** Haematoxilin-eosin (HE) staining and phospho-MEK and phospho-ERK immunohistochemistry of two PD3D lines OT326 and OT227 that are KRAS-wild-type and KRAS-mutant, respectively. Scale bars denote 100 μm for HE and immunohistochemistry. **b** CyTOF analysis of PD3Ds. Principal component analyses, colour-coded for EphB2, cleaved Caspase, phospho-MEK and phospho-ERK are shown. Red, yellow and blue colours of the scale represent high, intermediate and low signals, respectively. CyTOF data is available as a Source Data file

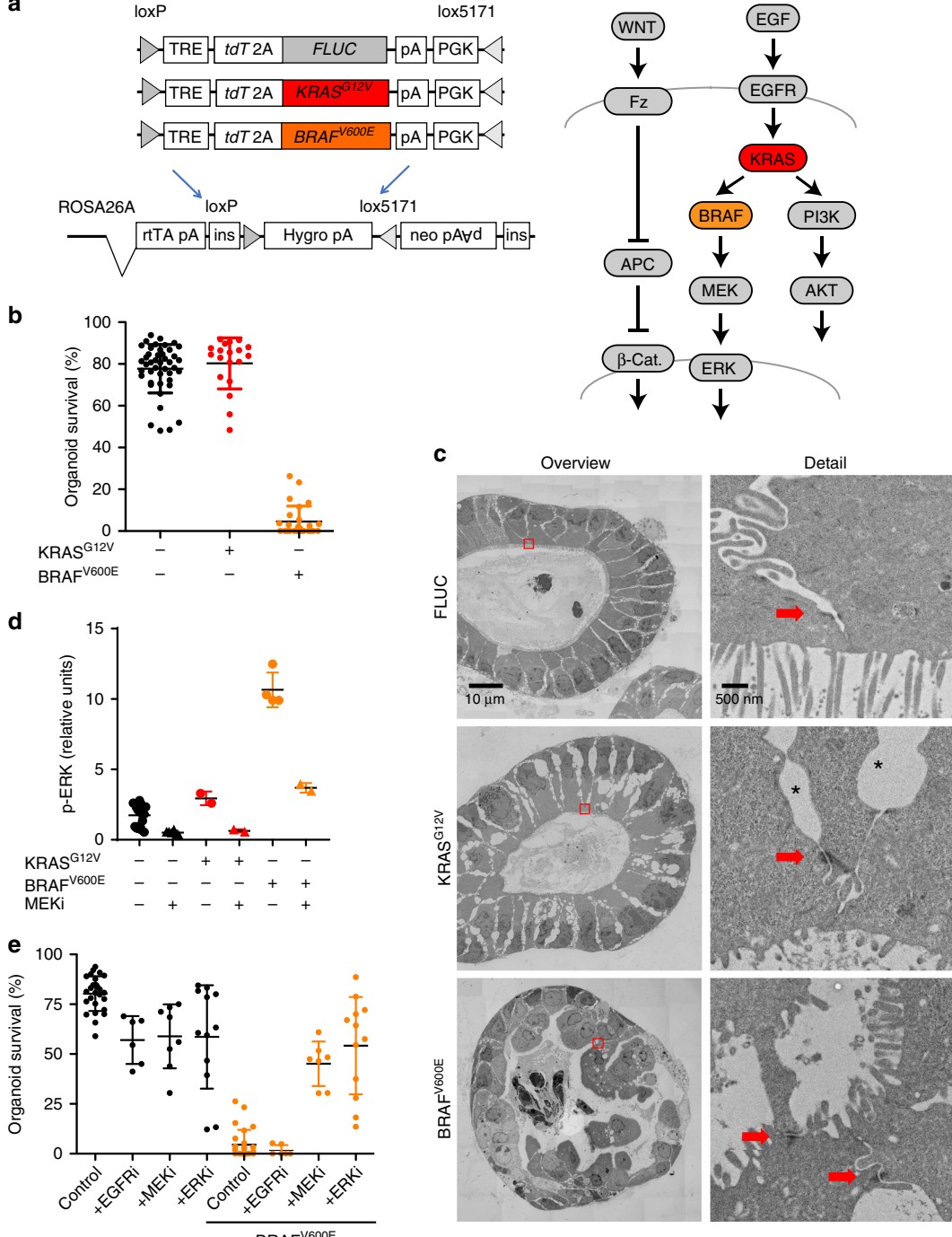

**Fig. 2** Transgenic BRAF$^{V600E}$, but not KRAS$^{G12V}$ disrupts organoids due to high ERK activity. **a** Simplified representations of transgenes and the RAS-ERK and Wnt/β-catenin pathways, indicating relative positions of the KRAS and BRAF proto-oncogenes. **b** Organoid survival 4 days after induction of oncogenic KRAS$^{G12V}$ or BRAF$^{V600E}$. Organoids are counted immediately after passaging, and fractions of surviving organoids were calculated at day 4. Control organoids comprise of mixed non-induced cultures of KRAS$^{G12V}$ and BRAF$^{V600E}$ lines. **c** Electron microscopy reveals loss of epithelial integrity after BRAF$^{V600E}$ induction. Images of the intestinal organoid epithelium, 24 h after induction of control FLUC, KRAS$^{G12V}$ or BRAFV$^{600E}$ transgenes. Detailed views (right) represent a zoom into areas marked by red boxes in the overviews (left). Detailed views show apical surfaces of adjacent enterocytes with brush border. Red arrows mark desmosomes. Intercellular vacuoles, most visible in the KRAS$^{G12V}$ model (marked by *) are likely fixation-induced artefacts, see ref. [27]. Scale bars are 10 μm in the overview panels and 500 nm in the detailed view panels. **d** Quantification of ERK phosphorylation in organoids, 24 h after induction of control, BRAF or KRAS transgenes, using a capillary protein analysis. **e** Quantification of organoid survival, 4 days after inhibition of EGFR, MEK, ERK and/or induction of BRAF$^{V600E}$, as in panel (**b**). Error bars in panels (**b**), (**d**) and (**e**) denote standard deviations. Data shown in panels (**b**), (**d**), and (**e**) are available as a Source Data file

clearly visible. In contrast, BRAF[V600E]-induced organoids displayed a continuous basal surface, whereas the apical side was grossly distorted, although it contained a brush border as evidence of polarisation. Nuclei were pleomorphic and no longer lined up basally but scattered at different positions. Cells were still attached to each other by desmosome bridges, indicating that the ongoing epithelial disorganisation was taking place in the presence of lateral cell adhesion.

To ascertain whether the epithelial disorganisation provoked by BRAF[V600E] was correlated with MAPK activity, we measured phosphorylation of ERK. We found that induction of BRAF[V600E], but not KRAS[G12V], resulted in high phospho-ERK levels in intestinal organoids, as determined by capillary protein analysis (Fig. 2d). BRAF[V600E]-induced organoid disintegration could be counteracted by inhibition of the BRAF-downstream MEK and ERK kinases using AZD6244/Selumetinib[31] and BVD-523/ Ulixertinib[32], respectively, but not by inhibition of the upstream EGFR tyrosine kinase receptor family using AZD8931/Sapitinib[33] (Fig. 2e), showing that the phenotype is due to excessive MEK-ERK activity. Indeed, only 24 h after BRAF[V600E] induction, almost all direct ERK target genes[34] were activated. In contrast, conditional expression of KRAS[G12V] induced RAS activity as measured by a RAS-GTP pull-down assay but had no obvious effect on bulk organoid transcription (Supplementary Fig. 1).

**BRAF[V600E] disrupts intestinal differentiation trajectories**. To uncover potential cellular heterogeneity in response to the oncogenes, we performed single-cell transcriptome analyses. We induced FLUC control, BRAF[V600E]- and KRAS[G12V]-transgenic organoids for 24 h, prepared single-cell suspensions, and stained them with a fluorescent antibody against the crypt cell marker CD44[35], and with a fluorescent dye to eliminate dead cells. Using single-cell sorting on the transgene-expressing organoids we next acquired samples of CD44-high crypt and CD44-low villus cells (see Supplementary Fig. 2 for FACS gating strategy), which were subjected to single-cell RNA sequencing. In total, we obtained transcriptomes of 167 cells with >1000 detected genes each, that were used for further analysis. Single-cell transcriptomes could be assigned to six interconnected clusters with help of k-means clustering and were visualised in a t-SNE-based representation (Fig. 3a). Mapping of signature genes for intestinal stem cells (ISCs), proliferative TA cells, differentiated enterocytes[20] and secretory Paneth cells[4], and the CD44 status as inferred from flow cytometry (Fig. 3b, c) confirmed the calculated differentiation trajectories (grey overlay in Fig. 3a): undifferentiated CD44-high ISC and TA cell signature genes were enriched in clusters 1 and 2 while Paneth cell marker genes were highest in cluster 2, indicating the differentiation route for secretory crypt cells; expression of enterocyte signature genes increased gradually in clusters 3–5, marking the CD44-low absorptive lineage.

We next considered the distribution of cells expressing specific transgenes (Fig. 3b): FLUC control and KRAS[G12V]-expressing cells intermingled throughout the clusters 1–5 of the normal cell differentiation trajectories. BRAF[V600E]-expressing cells were in contrast depleted from the central clusters 2–4, and instead formed outsider cluster 6, composed entirely of BRAF-induced cells. Notably, cells in cluster 6 uniformly expressed high levels of ERK target genes, regardless of whether they were sorted as CD44-high or CD44-low. Furthermore, cluster 6 cells also highly expressed *Anxa10*, which has been identified as a marker for BRAF-positive serrated adenoma[36] (Supplementary Fig. 3). The single-cell analysis thus showed that BRAF[V600E] imposed a specific gene expression programme onto intestinal cells, independent of their prior differentiation state. Transcriptomes of KRAS[G12V]-induced cells, in contrast, were undistinguishable

from FLUC control cells; however, we observed that KRAS[G12V]-induced cells showed a shift towards CD44-high undifferentiated cell types compared with FLUC controls (see Supplementary Fig. 2b).

**KRAS[G12V]-to-ERK signalling is cell type-specific**. To visualise ERK activity with single-cell resolution in organoids, we employed the Fra-1-based integrative reporter of ERK (FIRE) that translates ERK kinase activity into stability of a nuclear yellow-green venus fluorescent protein (Fig. 4a)[23]. FIRE fluorescence in organoids cultured in normal growth medium containing EGF was strong in crypts, whereas differentiated villus tissue was largely FIRE negative (Fig. 4b). In EGF-free medium, ERK activity in the crypt base persisted, likely due to autocrine and paracrine signals from EGF-producing Paneth cells[4].

We next conditionally expressed FLUC control, KRAS[G12V]-, or BRAF[V600E]-encoding transgenes in FIRE-transfected organoids (Fig. 4c). Transgene induction was often variable, as inferred by tdTomato fluorescence, allowing to compare individual tdTomato-positive cells with transgene-negative neighbouring tissue. tdTomato-FLUC control transgene expression had no influence on FIRE activity. In contrast, expression of KRAS[G12V] resulted in increased FIRE signals in crypt cells, which consistently displayed stronger reporter activity compared with adjacent KRAS[G12V]-negative cells. Surprisingly, a large majority of villus cells remained FIRE negative, despite strong tdTomato-KRAS[G12V] positivity. We confirmed the differential signal transduction from KRAS[G12V] to ERK using phospho-ERK immunohistochemistry (Fig. 4d). In line with our FIRE reporter data, p-ERK-positive cells were largely absent in central differentiated (Ki67-negative) villus areas of organoids, despite strong tdTomato-KRAS[G12V] staining. Taken together, our results show that ERK activity in differentiated villus epithelial cells can neither be increased by EGF in the medium nor by induction of oncogenic KRAS[G12V]. However, when we induced BRAF[V600E], we found widespread and strong FIRE signals across the complete organoid (Fig. 4c). This suggests a strict and cell type-specific control of signal transduction by oncogenic KRAS, but not BRAF, in intestinal epithelial cells.

Since FIRE fluorescence could distinguish cells responsive to KRAS[G12V], we next used the reporter to assist selection of cells for single-cell RNA sequencing. Our aim was to define cell types with high ERK activity, either in response to KRAS[G12V] or as part of the normal cell hierarchy. For this, we induced organoids with the integrated ERK reporter for KRAS[G12V] or FLUC, prepared single-cell suspensions and sorted cells by FACS into 96-well plates for transcriptome analysis (see Supplementary Fig. 4 for FACS gating strategy). We focussed on single cells with high transgene (tdTomato) signal that were either positive or negative for FIRE (venus) fluorescence (Fig. 5a). In total, we obtained 197 single-cell transcriptomes. K-means clustering into eight groups and t-SNE-based visualisation revealed the cell type distribution (Fig. 5b, c). Cluster 1 was enriched for undifferentiated crypt (ISC and TA) marker genes, whereas clusters 2–4 were defined by Paneth cell signature genes (Fig. 5d; Supplementary Fig. 5). Cluster 2 was enriched for Paneth cell markers such as *Lyz1*, encoding Lysozyme[37], and several genes encoding Defensins, while other cluster-defining genes such as *Mptx1* and *Agr2* in cluster 4 hint at a high degree of Paneth cell heterogeneity. Clusters 5–8 formed a differentiation trajectory for absorptive cells, with *Ifabp1* as the top defining gene for clusters 5–7 (Supplementary Fig. 5).

Using this information, we assessed the distribution of transcriptomes derived from KRAS[G12V]-induced FIRE-high cells (Fig. 5c, d). These were confined to distinct aggregates

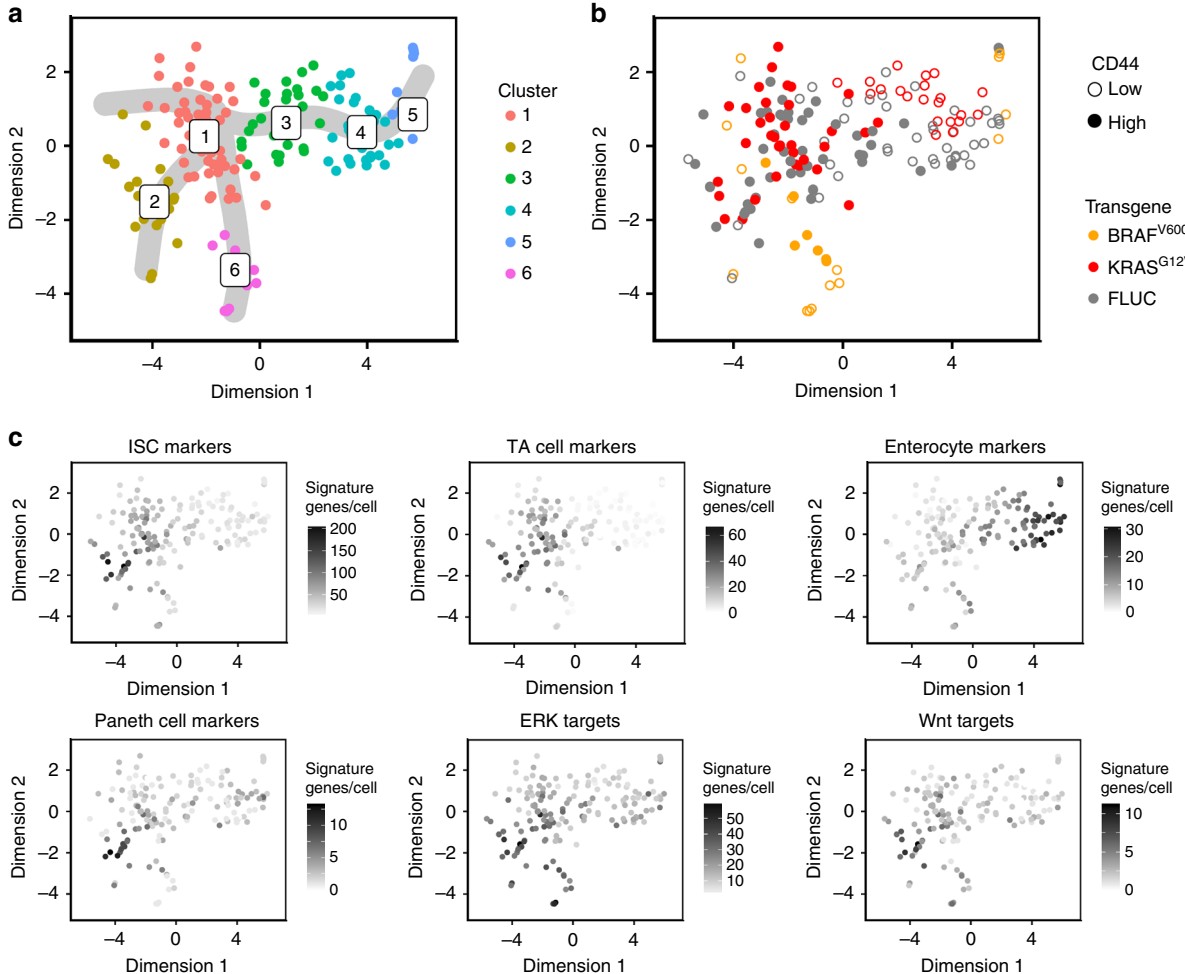

**Fig. 3** Differential effects of BRAF[V600E] or KRAS[G12V] on gene expression and intestinal cell hierarchies. All panels: t-SNE visualisations and clustering of organoid single-cell transcriptomes clustered with k-means, 24 h after induction of FLUC control, BRAF[V600E] or KRAS[G12V] transgenes. **a** Colour code for six k-means clusters, and inferred differentiation trajectories starting at cluster 1 shown as grey overlay. **b** Colour code for transgene and CD44 positivity, as inferred from flow cytometry. CD44 positivity was used to direct cell selection, and thus relative fractions of CD44-high and -low cells are not representative. For CD44 status of the cell populations, see Supplementary Fig. 2. **c** Mapping of cell- and pathway-specific differentiation signatures. Numbers of signature genes detected are given per single-cell transcriptome

encompassing the undifferentiated cell zone of cluster 1, as well as transcriptomes inhabiting the outer right rim of the t-SNE representation that we above assigned to be derived from late-stage enterocytes and Paneth cells. Immunofluorescence microscopy using the Paneth cell marker Lysozyme confirmed high FIRE activity in this cell type after KRAS[G12V] induction (Supplementary Fig. 6). In contrast, a central area of the t-SNE plot encompassing the largest clusters 5 and 6 of bulk enterocytes was almost devoid of KRAS[G12]-producing FIRE-high cells but harboured many KRAS[G12V]/FIRE-low cells, confirming that enterocytes generally cannot activate ERK, even when expressing oncogenic KRAS[G12V]; however, a specific subset of presumably late-stage enterocytes displayed high ERK activity.

**KRAS[G12V] interacts with GSK3β inhibition**. In order to understand how β-catenin- and MAPK-networks interact in controlling cell differentiation and ERK phosphorylation in intestinal epithelium, we performed a network perturbation study using kinase inhibitors, followed by mass cytometry in KRAS[G12V]-inducible and FLUC control organoids. For this, we induced the transgenes in 3-day-old organoids, subsequently treated them with an GSK3β inhibitor (CHIR99021) for 24 h to stabilise β-catenin[38], and used MEK and p38 inhibitors

(AZD6244 and LY2228820/Ralimetinib[39], respectively) for 3 h to inhibit key kinases as part of the intestinal cell signalling network (Fig. 6a). We measured a total of 160,000 transgene-positive cells, representing 12 multiplexed samples.

To discern the immediate effects of KRAS[G12V] and stabilised β-catenin on intestinal cell hierarchies, we assessed the distribution of cell type markers (Fig. 6b). As a positive control for the effect of GSK3β inhibition on β-catenin activity, treatment with CHIR99021 increased levels of the β-catenin target protein Axin2[40]. We observed that both induction of KRAS[G12V] and treatment with the GSK3β inhibitor, increased median levels of crypt cell markers EphB2, CD44 and CD24, and for all three proteins, KRAS[G12V]-induced cells that were additionally treated with the GSK3β inhibitor had the highest levels. These results are in line with prior evidence that oncogenic KRAS and β-catenin activities can inhibit or reverse intestinal cell differentiation and provide clonal benefits linked to crypt cell fate[41,42].

We used k-means clustering to allocate KRAS[G12V]-induced cells to six clusters defined by levels of cell type and surface markers CD24, CD44, EphB2, Krt20 and apoptosis marker cleaved Caspase 3 (Fig. 6c). p-ERK-positive cells were enriched in clusters 5 and 6, while cleaved Caspase 3-positive cells were found in Cluster 5 (Fig. 6d, e). Based on gradual loss of the crypt cell

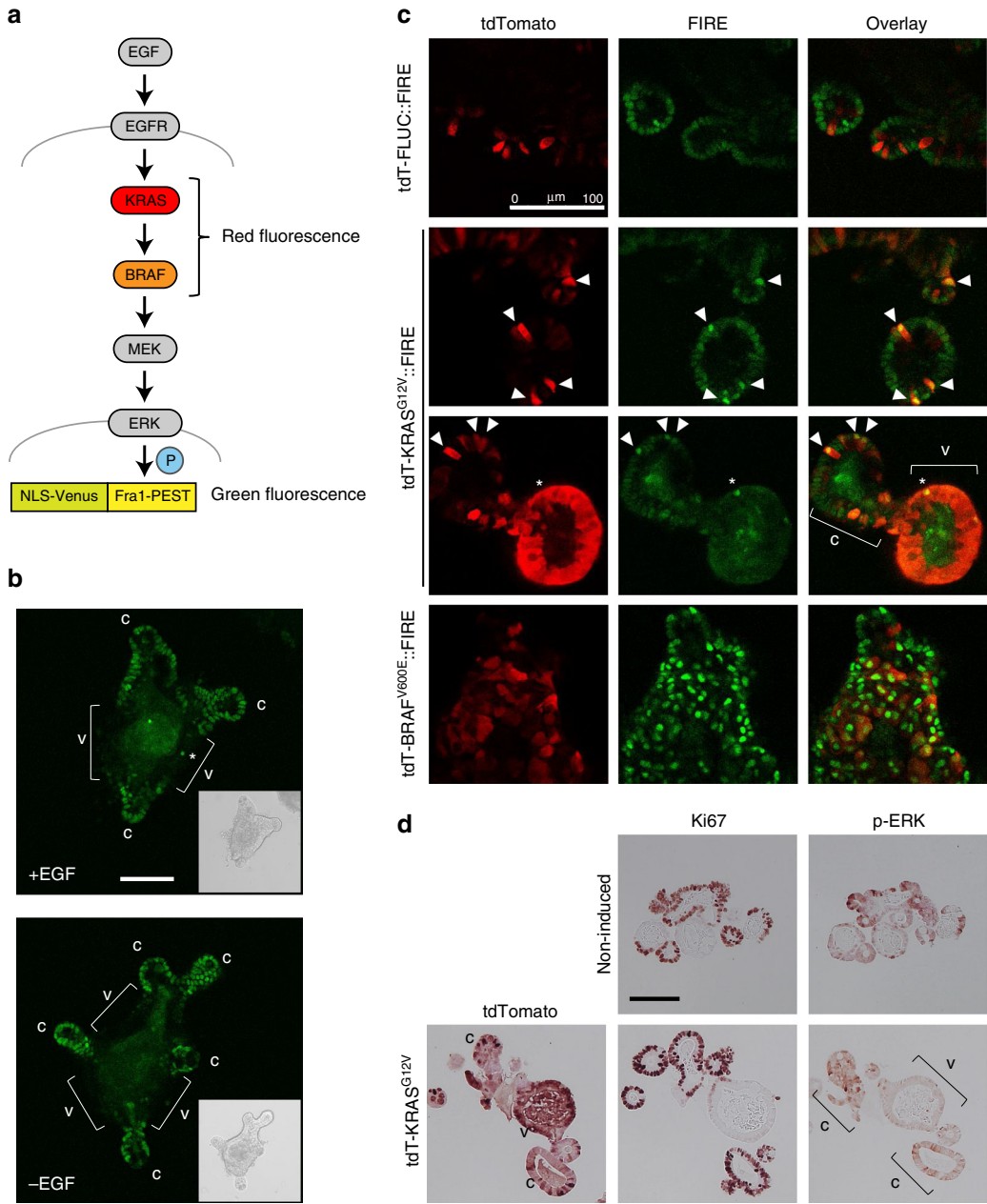

**Fig. 4** Visualisation of ERK activity by FIRE reveals KRAS^G12V-responsive cells. **a** Schematic representation of signalling pathway and reporter. **b** FIRE activity in wild-type intestinal organoids, in the presence and absence of EGF in the culture medium, as indicated. Asterisk marks isolated FIRE-high villus cell. **c** Fluorescence microscopy images showing transgene expression (red), FIRE activity (green), and overlays in intestinal organoids, taken 2 days (FLUC, KRAS) or 1 day (BRAF) after transgene induction. Arrow heads mark KRAS^G12V/FIRE-high crypt cells, asterisk marks FIRE-high villus cell, respectively. **d** Immunohistochemistry of tdTomato, Ki67 and p-ERK in intestinal organoids, as indicated. In panels (**c**) and (**d**), c and v demarcate crypt and villus areas, respectively. Scale bars are 100 μm in all panels

markers EphB2, CD44 and CD24, we concluded that clusters 1–4 represent a crypt-to-villus gradient, interconnecting with cluster 5 at the end of the differentiation trajectory (Fig. 6f). Clusters 3 and 4 had the lowest phospho-MEK and phospho-ERK levels, and also contained lowest levels of the Wnt/β-catenin target Axin2, in agreement with differentiated villus cell status. High levels of CD24 marked p-ERK-positive cells in cluster 6 as presumptive Paneth cells[4]. Interestingly, we observed that fractions of cells allocated to the clusters were modified by both KRAS^G12V induction and GSK3β inhibition (Fig. 6g): both treatments increased the percentage of cells in cluster 1, representing the presumptive undifferentiated crypt cells, as inferred from high levels of markers such as EphB2 and CD44[1,20]. The combination

of KRAS^G12V and GSK3β inhibition had the greatest effect and furthermore strongly decreased the fraction of cells in the apoptotic cell cluster 5. These data suggest that oncogenic KRAS and β-catenin stabilisation can both favour crypt cell fate over differentiation, at least on the level of cell type marker expression.

**Cell type-specific differences in ERK feedback regulation**. To quantitatively dissect differences in signalling in the cell types, we assigned the cells measured under different perturbed conditions to the six clusters defined above according to their shortest Euclidian distance (Fig. 6c). For each condition and each cluster, we calculated average phosphorylation levels of MEK, ERK,

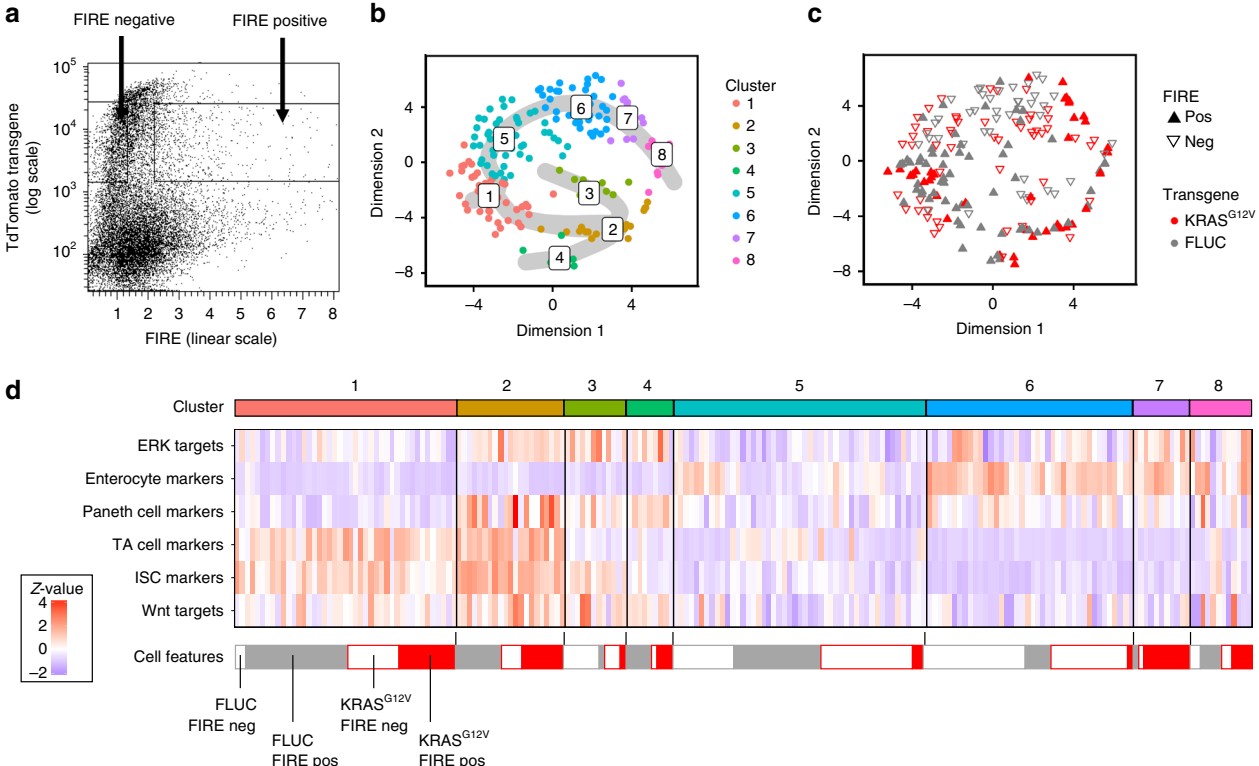

**Fig. 5** Single-cell RNA sequencing reveals KRAS^G12V-responsive and -unresponsive organoid cells. **a** Fluorescence-activated cell sort gates for FIRE-negative and -positive cells. **b** t-SNE visualisation colour-coded for eight clusters identified with k-means clustering. Differentiation trajectories starting at cluster 1 are shown as grey overlay. **c** t-SNE visualisation displaying colour codes for transgene and FIRE positivity. Filled upward-pointing triangles: FIRE-high; outlined downward-pointing triangles: FIRE-low. Red: KRAS^G12V; grey: FLUC. **d** Heatmap of z-transformed signature scores per cell for cluster cell type identification. Signature scores correspond to the number of expressed signature genes per cell normalised to gene detection rate and signature length. Blue: low target gene signature abundance; Red: high target gene signature abundance. Cluster colour codes are given above, and transgene and FIRE positivity codes are given below the heatmap

4EBP1, p38, ribosomal protein S6 and total protein levels of IκBα and Axin2, forming a Wnt-, MAPK-, NFκB- and mTOR network (Fig. 7a, b). Cellular signalling states varied strongly between the cell clusters 1 and 6. However, comparison between FLUC control and KRAS^G12V-expressing organoid cells showed that signalling within each cell type was very comparable, except for an increase in phosphorylation levels of MEK and ERK in clusters 5 and 6 in cells expressing KRAS^G12V. This was in contrast to results from similar CyTOF experiments performed with the BRAF^V600E-inducible organoid line, which resulted in high MEK and ERK phosphorylation across all cell type clusters (Supplementary Fig. 7).

We next employed network modelling using Modular Response Analysis (MRA)[43,44]. This approach allows to quantify signal transmission from perturbation-response data and, by using likelihood ratio tests, to pinpoint which signalling routes are different between the clusters. The method requires fold changes in network node activity after perturbation (calculated from data shown in Fig. 7a) and a literature-derived network (as shown in Fig. 7b) as input and calculates so-called response coefficients that reflect the strength of signalling interactions. When we applied this modelling framework to the signalling perturbation data of the six clusters for the KRAS-mutant organoid data, we observed that only 5 out of 11 signalling routes significantly differed between clusters with two of them constituting the feed-forward and feedback signalling paths between MEK and ERK (Fig. 7b, c). In clusters 5 and 6, which were the clusters that showed elevated phosphorylation levels of MEK and ERK levels after KRAS^G12V induction, the model

unveiled that these cells enable RAS-ERK signalling by two mechanisms: signal transmission from MEK to ERK was enhanced, while ERK-dependent negative feedback inhibition of RAF and upstream components was attenuated when compared with the other clusters. In contrast, clusters 3 and 4, which exhibited the lowest phospho-ERK levels that were also not increased by KRAS^G12V, MEK-to-ERK feed forward signal transduction was low, and this coincided with strong ERK-dependent feedback inhibition. Cluster 1, which based on surface marker expression represents undifferentiated crypt cells, had strong feedback inhibition according to the model. This cluster had intermediate phospho-ERK levels as measured by CyTOF (Fig. 6f) and was FIRE positive (Fig. 4c). However, phospho-ERK levels were unresponsive to KRAS^G12V in this cluster and our model predicts that this could be due to strong ERK feedback.

To model differences between KRAS^G12V-induced and control cells within each cluster, we employed comparative MRA modelling that resulted in cluster-specific signalling models that consider the influence of KRAS^G12V on the signalling network per cluster (Fig. 7d). We could discern only few differences. Most importantly, KRAS^G12V enhanced signalling from MEK to ERK in clusters 5 and 6. Furthermore, we observed that KRAS^G12V modulated the effect of Wnt/Axin2 signalling on mTOR in cluster 2.

As our functional studies and the modelling showed that RAS-ERK signal transduction can be differently wired between cell types, we tested whether this was also true for CRC cell lines. Indeed, when we compared ERK phosphorylation in response to transfected KRAS^G12V in SW48 and Caco2 CRC cells (that have

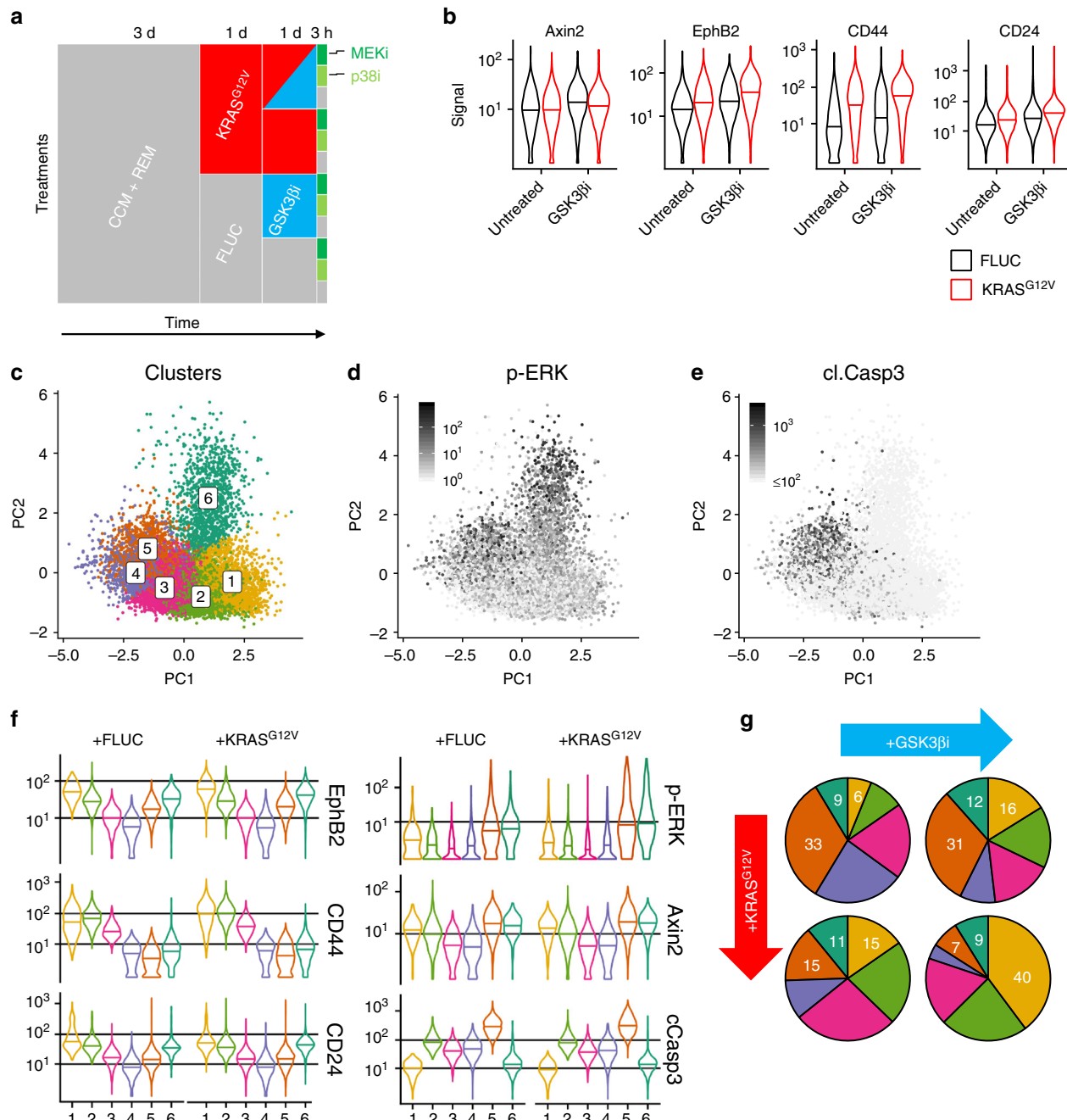

**Fig. 6** CyTOF analysis reveals KRAS^G12V- and GSK3β inhibitor-responsive p-ERK high cell clusters. **a** Schematics for generation of network perturbation data by CyTOF. In short, organoids were established from KRAS^G12V- and FLUC transgenic mice, induced for transgene expression after 3 days, and treated with GSK3β inhibitor for 1 day and with MEK and p38 inhibitors for 3 h before harvesting. Finally, 12 samples were subjected to multiplexed CyTOF analysis. **b** Distributions of cell type markers in organoid cells induced for FLUC or KRAS^G12V transgenes plus/minus GSK3β inhibitor treatment. Central lines of violin plots denote median values. **c** PCA showing colour code of k-means clustering in KRAS^G12V-induced cells by EphB2, CD44, CD24, Krt20 and cleaved Caspase 3 signal strength. **d**, **e** Mapping of signal strength for p-ERK and cleaved Caspase 3 on PCA, as in (**c**). **f** Distribution of EphB2, CD44, CD24, Axin2, p-ERK and cleaved Caspase 3 signals in clusters 1–6, as above. Central lines of violin plots denote median values. **g** Fractions of cells in clusters 1–6, in organoid cells induced for FLUC or KRAS^G12V transgenes plus/minus GSK3β inhibitor treatment. Numbers denote percentages of cells in clusters 1, 5, 6. CyTOF data are available as a Source Data file

no mutations in KRAS, NRAS or BRAF), we found that Caco2 cells were KRAS^G12V-responsive, while SW48 cells were KRAS^G12V-insensitive, extending a recent study that shows only subtle effects of RAS mutants in SW48[45] (Supplementary Fig. 8).

Our cluster-specific signalling data (Fig. 7a) showed correlated activities of RAS-ERK, Wnt/β-catenin and other signalling pathways, as they were generally higher in presumptive crypt cell clusters 1 and 6, but lower in the presumptive villus enterocyte clusters 3 and 4. Contrarily, the MRA approach suggested that Axin2 as read-out of Wnt/β-catenin signalling was a negative regulator of RAS-ERK. We reasoned that, as we stimulated Wnt/β-catenin signalling by inhibiting GSK3β, attenuation of RAS-ERK could be caused by other targets of GSK3β. To more directly assess the effect of β-catenin

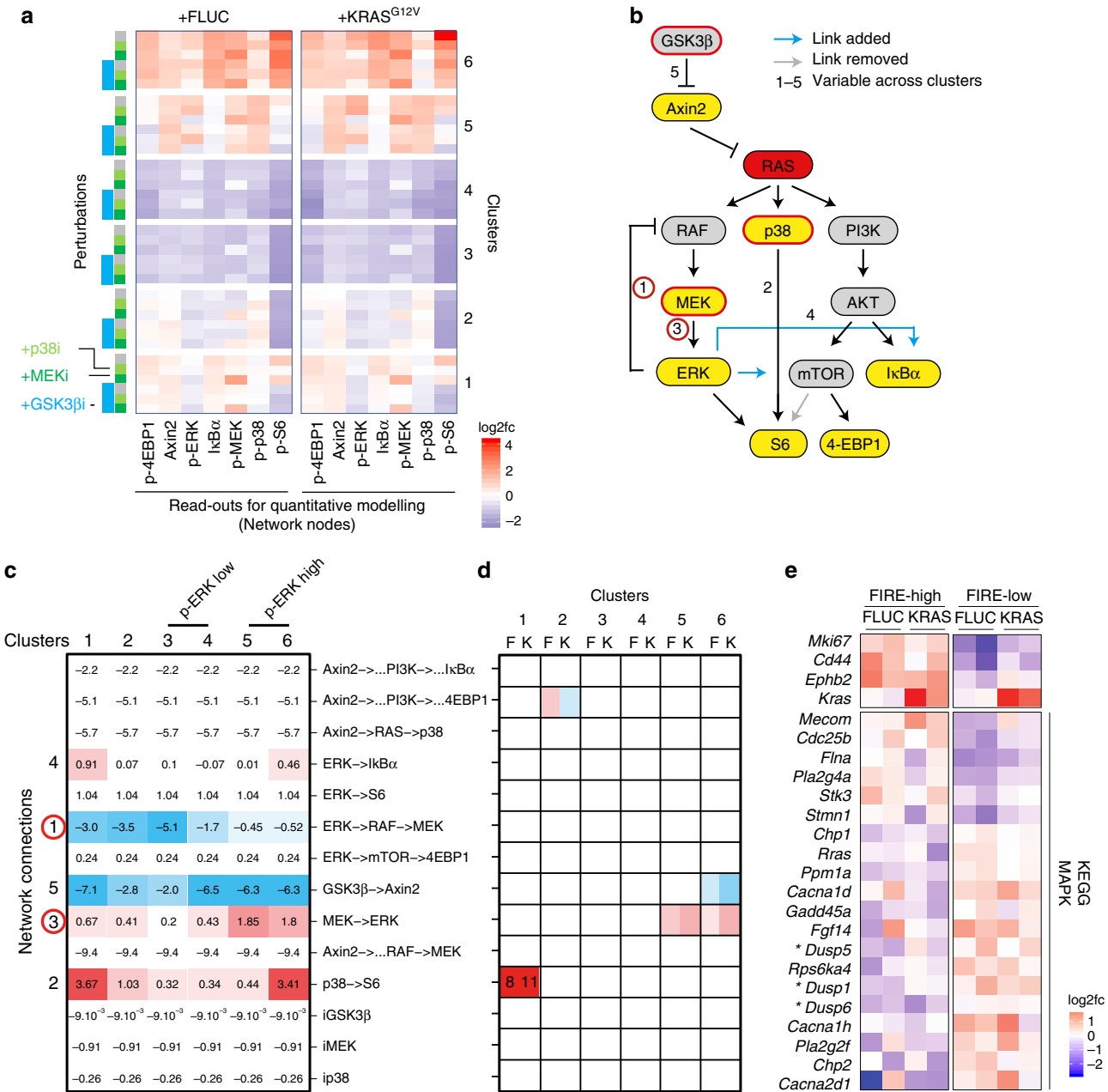

**Fig. 7** Network quantification identifies cell type-specific differences in KRAS to ERK signalling. **a** Protein phosphorylation and abundance CyTOF data by treatment and cell clusters, as in Fig. 6c. Log2 fold changes to average untreated FLUC-induced control line are given. **b** Signalling network structure used for modelling. The network was re-parametrised from a starting network, using the experimental data to remove and add connections, denoted by grey and blue arrows, respectively. **c** Signalling quantification of identifiable network links using Modular Response Analysis. Numbers 1–5 in panels (**b**) and (**c**) show network connections with significant differences between clusters in order of detection. Red circles mark MEK-ERK and ERK-MEK connections identified as having different strengths in clusters with high vs. low ERK phosphorylation after KRAS^G12V induction. **d** MRA modelling of differences between KRAS^G12V-induced and FLUC control cells within each cluster. K and F mark KRAS^G12V and FLUC control cluster pairs, respectively. Cluster pairs exhibiting KRAS^G12V-specific differences are shown in red and blue, indicating regulation strengths. **e** Colour-coded gene expression data from cells sorted by high and low FIRE activity, as indicated. Upper panel shows marker genes (*Mki67*, encoding Ki67, for proliferative cells, *Cd44* and *Ephb2* for crypt cells and *Kras*), lower panel shows 20 significantly regulated genes between the conditions. In total, 269 genes encoding MAPK network components in KEGG were tested. Asterisks indicate dual-specificity phosphatases

on RAS-ERK, we therefore performed further experiments in transgenic organoids in which we induced transgenic stabilised β-catenin or withdrew Wnt ligands. The data show that supplementation and abrogation of β-catenin activity both result in lower ERK phosphorylation in organoids (Supplementary Fig. 9).

As the results of the modelling pinpointed differences in signal transduction from KRAS^G12V to ERK to feed-forward and feedback signalling between MEK and ERK, we investigated which molecular mechanism might attenuate ERK activation in intestinal cells. For this we sorted FLUC control or KRAS^G12V-induced organoids with respect to their FIRE reporter levels, as

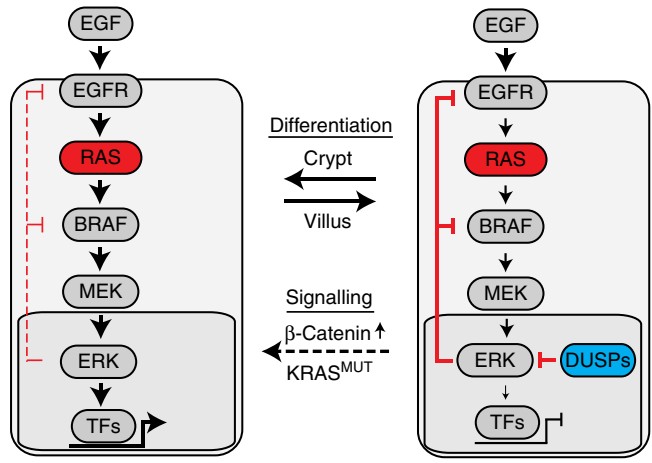

**Fig. 8** Model of cell type-specific regulation of ERK activity. ERK is regulated cell type-specific and cell-intrinsic via different strengths of feedback inhibition and feed-forward signalling from MEK to ERK. Dual-specificity phosphatases (DUSPs) are important regulators of ERK activity. β-catenin and KRAS[G12V] activities modulate cell fate decisions towards the generation of cells with high ERK activity, likely in part due to low expression of genes encoding DUSPs

above (Fig. 5a) and performed bulk low-input RNA sequencing. When inspecting 267 genes whose products are implicated in the MAPK signalling pathway, we noticed three dual-specificity phosphatases among a total of 20 differentially expressed genes (Fig. 7e). As these phosphatases are known to dephosphorylate ERK[46], we consider the DUSP1, DUSP5 and DUSP6 gene products as candidate mediators of attenuated MEK-ERK signal transmission that we observe in the differentiated cells of clusters 3–4.

## Discussion

The mass cytometry, reporter assay and single-cell RNA sequencing data that we present here support a model of cell type-specific and cell-intrinsic regulation of the terminal MAP kinase ERK (Fig. 8). We found that oncogenic KRAS can activate ERK only in specific cell types of the intestinal epithelium, while other cells such as enterocytes were insensitive to ERK activation by oncogenic KRAS. Our quantitative network model suggested that ERK activity is attenuated in the latter cells by increased feedback inhibition and reduced feed-forward signalling from MEK to ERK. In agreement, transcriptome analysis showed that dual-specificity phosphatases (DUSPs) are selectively expressed in cells with low ERK activity and may therefore contribute to cell type-specific suppression of ERK. In addition, crypt cells expanded in organoids upon induction of KRAS[G12V], and this effect was increased when KRAS[G12V] was combined with GSK3β-mediated β-catenin activity.

The unexpected disparities in the levels of KRAS[G12V]-induced ERK phosphorylation in different cell types derived from the intestinal epithelium extend our understanding of how KRAS, the most prevalent oncogene in CRC, exerts its effects. It is of note that our study only analysed the G12V mutation of KRAS, and other mutations such as the more common G12D variant or mutations at amino acids 13, 61 or 146 may engage different effectors[47,48]. Local differences of ERK activity have recently been found in clinical specimens of CRC, including KRAS-mutant CRC[14]. In this previous study, ERK levels were generally higher in cancer cells adjacent to stromal cells at the invasive front, and lower in more central areas of cancer specimens, in line with

modulation of ERK activity by cues from the microenvironment. Our results agree with a model of dynamic ERK activity in cancer tissues, and we find in addition that ERK is not only regulated via external cues from the microenvironment, but also by the cell-intrinsic differentiation state.

Quantitative modelling of ERK activity from perturbation data revealed that the markedly distinct abilities of KRAS[G12V] to activate ERK are due to different strengths of two network connections between cell types, namely MEK to ERK feed forward and ERK to MEK negative feedback signalling. Negative feedback within the pathway is well-known, and has been linked to induction of cellular senescence[49] and to therapy resistance in CRC[44,50]. Our study has identified dual-specificity phosphatases as candidate genes modulating MEK-to-ERK signalling during intestinal differentiation. Indeed, the *Dusp1*, *Dusp5* and *Dusp6* genes we have identified to be differentially expressed between RAS-to-ERK-responsive and -insensitive intestinal cells, encode ERK-specific phosphatases[46]. A protein phosphatase network, including DUSPs, has recently been found to control ERK activity and differentiation in skin[51]. Our study suggests the existence of similar control mechanisms in the intestine. Dual-specificity phosphatases targeting ERK have been implicated in resistance of non-small lung cancer to anti-EGFR therapy[52]. It remains to be seen whether differences in RAS-ERK signalling beyond the mutational status of BRAF, NRAS and KRAS have prognostic value in CRC. Today, CRC patients with wild-type status of the predictive markers KRAS, NRAS and BRAF are eligible for anti-EGFR therapy[53,54]. However, the treatment also generates mixed outcomes among eligible patients, showing the need to identify novel markers and mechanisms contributing to differences in EGFR-RAS-ERK signal transduction and therapy response.

Paneth cells and enterocytes represent the main differentiated cell types of the intestinal epithelium, and we found that they have marked differences in their abilities to activate ERK. It is of note that the former reside in the Wnt/β-catenin-high crypt compartment, while the latter inhabit a compartment with low β-catenin activity. Indeed, a functional role for Wnt/β-catenin in the activation of ERK in intestinal epithelium has been proposed before[55]. In our experiments using organoids, p-ERK levels were lowered when we increased or decreased β-catenin activity (Supplementary Fig. 9A, B). We hypothesise that this is due to a loss of all crypt cells or of phospho-ERK-high Paneth cells in the models with low and high β-catenin activity, respectively. Our data agree with a recent publication proposing negative interaction between Wnt/β-catenin and ERK signals in intestinal stem cells[56]. Indeed, activation of β-catenin by treatment with a GSK3β inhibitor increased the fraction of cells positive for stem cell marker CD44, but not levels of MEK or ERK phosphorylation per cell in the CyTOF experiments.

ERK activity is regulated on multiple levels, including its subcellular localisation[57]. We used different approaches to quantify ERK activity. While mass cytometry and capillary protein analysis measure total levels of ERK phosphorylation, the FIRE reporter assesses nuclear ERK activity only[23]. These activity measures for crypt cells suggest divergence of total and nuclear activity. While all crypt cells appeared FIRE positive, the presumptive undifferentiated crypt cell cluster 1 showed only intermediate levels of ERK phosphorylation in the CyTOF analysis. Further analyses will be required to understand how subcellular ERK activity is controlled in a cell type-specific manner.

In our experiments, signal transduction from BRAF[V600E] to ERK was relatively independent of cellular context. Extending previous studies[29,30], we found that high levels of ERK activity induced by oncogenic BRAF[V600E] are not tolerated in the intestine. This is in contrast to CRC and cell lines, where BRAF[V600E] amplifications exist and are selected for by MEK inhibition[58,59]. It

thus appears that the corridor for acceptable ERK activity is tuneable during CRC progression and under selective pressure exerted, for instance, by targeted therapy. Therefore, our findings are reminiscent of the "just right" signalling model that has been proposed to explain step-wise increases of β-catenin activity in CRC progression[60].

## Methods

**Generation of transgenic mice.** Transgene cassettes were constructed by linking tdTomato to human BRAF(V600E), KRAS(G12V) and/or murine stabilised mutant Ctnnb1 (S33A, S37A, T41A, S45A) or firefly luciferase via 2A peptides, and subsequent cloning of these gene combinations into a doxycycline-inducible expression cassette flanked by heterologous loxP sites, and integrated into a previously modified Gt(ROSA)26Sor locus of F1 hybrid B6/129S6 embryonic stem cells by Cre recombinase-mediated cassette exchange[28]. Transgenic animal experiments were approved by Berlin authorities LAGeSo (G0185/09, G0143/14).

**Organoid and cell line culture.** Mouse organoid cultures were initiated and propagated as described before[27], using 48-well plates with 15 μl droplets of Matrigel (Corning) per well overlaid with 300 μl crypt culture medium containing EGF (50 ng/ml), Noggin (100 ng/ml) and R-Spondin1 (functionally tested from R-Spondin-conditioned medium; CCM-REN). Transgenes were induced by addition of 2 μg/ml doxycycline to the medium.

To obtain adenomatous organoids (spheroids) after induction of stabilised β-catenin, R-Spondin was removed after induction of the transgene encoding β-catenin[stab] alone or in combination with KRAS[G12V]. Spheroids were dissociated with TrypLE (Gibco) for 3 min and Rho kinase inhibitor Y27632 (10 μM) was added to the culture medium after passaging to prevent anoikis.

For viral transfection, a protocol from reference[61] was employed, with modifications: organoids were cultured in the presence of Y27632 and the GSK3β inhibitor CHIR99021 for 2 days. Next, organoids were disaggregated into single cells using TrypLE (Gibco) for 5 min at 37 °C. Cell suspensions were spin-occulated in an ultra-low adhesion round bottom 96-well plate with the virus at 300 g for 45 min. Subsequently, cells were resuspended in Matrigel, and cultured for 2 days in CCM-REN supplemented with Y27632 and CHIR99021. Medium was then replaced by CCM-REN containing 2 μg/ml puromycin to select for transfected cells. As viral transfection initially resulted in organoid pools that were heterogeneous for FIRE reporter activity, single FIRE positive organoids were manually selected and propagated before experimental analysis.

Organoid survival was scored as follows: cultures were passaged, inhibitors and doxycycline were applied with the culture medium directly after passaging. Individual wells were imaged using the z-stack function of BioZero observation and analyser software (Keyence) on day 1 and 4 (organoids) or day 1 and 6 (spheroids), and full focus reconstructed images were used for quantification.

Patient-derived organoids (PD3Ds) were obtained from the biobank of the Charité - Universitätsmedizin Berlin and experiments were approved by the ethics commission of Charité - Universitätsmedizin Berlin (EA1/011/18). Cells were cultivated in 24-well plates (Corning) in medium containing human bFGF (20 ng/ml, Sigma Aldrich) and EGF (50 ng/ml, Sigma Aldrich), as published[62].

SW48 and Caco2 CRC cells were cultured in L-15 and DMEM, respectively, supplemented with 10% foetal bovine serum. Cells were transfected using Lipofectamin 3000 Transfection Reagent (Thermo Fischer) with vectors encoding BRAF[V600E], KRAS[G12V] or FLUC linked to tdTomato and the pTet-on Advanced Vector (Clontech). Cells were starved in medium containing 0.1% foetal bovine serum and induced with 2 μg/ml doxycycline 48 h after transfection. Twenty-four hours later, cells were harvested using TrypLE, washed, rested for 30 min at 37 °C in starvation medium and fixed in 4% paraformaldehyde (PFA) for 15 min at 37 °C. Fixed cells were washed in PBS/1% bovine serum albumin (BSA), permeabilised in MeOH at −20 °C overnight, and immunostained with Alexa Fluor 488 mouse anti-ERK1/2 (pT202/pY204) antibody (1:10; 612592, BD Bioscience) for 30 min. Cell lines are routinely checked for mycoplasma contamination and panel-sequenced for oncogenic mutations to confirm identity.

The following inhibitors (SelleckChem) were employed: AZD6244 (10 μM), BVD-523 (3 μM), AZD8931 (50 nM), CHIR99021 (6 μM), Gefitinib (1 μM) and LY2228820 (200 nM).

**Mass cytometry (CyTOF).** For CyTOF analysis, we used the following pre-conjugated antibodies (Fluidigm) as per manufacturers recommendation: CD24 (for mouse: 150-Nd, 3150009B, for human: 169-Tm, 3169004B), CD44 (for mouse: 162-Dy, 3162030B, for human: 166-Er, 3166001B), cleaved Casp3 (142-Nd, 3142004A), p-H2AX [S139] (147-Sm, 3147016A), p-Akt [S473] (152-Sm, 3152005A), p-p38 [T180/Y182] (156-Gd, 3156002A), Ki67 (162-Dy, 3168007B), IκBα (164-Dy, 3164004A), p-ERK1/2 [T202/Y204] (171-Yb, 3171010A) and p-S6 [S235/236] (175-Lu, 3175009A).

For antibodies not available as metal-conjugates, we used the Maxpar Antibody Labeling Kit (Fluidigm) according to manufacturer's instructions for addition of the respective metal tags: Axin2 (145-Nd, Abcam, ab32197, 2 μg/ml), p-MEK1/2

[S217/221] (151-Eu, CST, 41G9, 2 μg/ml), EphB2 (158-Gd, BD, 2H9, 2 μg/ml), p-4e-BP1 [T37/46] (170-Er, CST, 236B4, 2 μg/ml) and Krt20 (176-Yb, CST, D9Z1Z, 2 μg/ml). The yield after conjugation was determined using a NanoDrop spectrometer measuring the absorbance at 280 nm wavelength.

For measurements, organoids were harvested in PBS and digested to a single-cell solution in 1:1 Accutase (Biolegend) and TrypLE (Gibco) with addition of 100 U/ml Universal Nuclease (Thermo Scientific) at 37 °C. Cells were counted and a maximum of 500,000 cells were stained with 5 μM Cell-ID Cisplatin (Fluidigm) in PBS for 5 min at 37 °C. After washing, cells were resuspended in their respective growth medium and allowed to rest for 30 min at 37 °C. Subsequently, cells were resuspended in BSA/PBS solution, mixed 1:1.4 with Proteomics Stabilizer (Smart Tube Inc.) and incubated for 10 min at room temperature. Afterwards the cells were frozen at −80 °C for storage.

One day prior to analysis, cells were thawed in a 37 °C water bath and mixed with Maxpar Cell Staining Buffer (CSB, Fluidigm). We used the Cell-ID 20-Plex Pd Barcoding Kit (Fluidigm) to label different samples and performed a downscaled version of the manufacturer's recommended protocol. Cells were washed again in CSB, then in Barcode Perm Buffer (Fluidigm). After resuspension in 200 μl Barcode Perm Buffer, 25 μl of the diluted Barcoding Reagents were added to the respective samples and incubated for 30 min at room temperature. Afterwards cells were washed twice in CSB, pooled into one tube and counted. In total, 3 × 10^6 cells were then stained with a surface antibody cocktail for 30 min at room temperature. After washing in CSB, cells were refixed in 2% methanol-free formaldehyde solution (Pierce; diluted in Maxpar PBS, Fluidigm) for 10 min at room temperature. Cells were washed in CSB and put on ice for 10 min. Next, cells were permeabilised by adding 4 °C methanol for 15 min. Cells were washed twice in CSB and incubated with a phospho-protein antibody cocktail for 30 min at room temperature. Cells were washed twice in CSB and incubated with 62.5 nM Cell-ID Intercalator-Ir in Maxpar PBS for 20 min at room temperature. Cells were washed in Maxpar PBS and fixed in 2% methanol-free formaldehyde overnight at 4 °C. The day after, cells were washed with CSB and then twice with Milli-Q water. Cell number was adjusted to 2.5–5 × 10^5/ml with Milli-Q water, cells were filtered through a 20 μm cell strainer (CellTrics, Sysmex) and supplemented 1:10 with EQ Four Element Calibration Beads (Fluidigm). Data were acquired on a Helios CyTOF system. Mass cytometry data were normalised using the Helios software and bead-related events were removed. Doublets were excluded by filtering for DNA content (191Ir and 193Ir) vs. event length, and apoptotic debris removed by a filter in the platin channel (195Pt). De-convolution of the barcoded sample was performed using the CATALYST R package version 1.5.3[63].

**Capillary protein quantification and RAS activity assay.** Protein sample preparation and quantification of p-ERK was performed using a WES capillary system (12–230 kD Master kit α-Rabbit–HRP; PS-MK01; Protein Simple) and the antibody p-ERK/2(T202/Y204) (1:50; #9101, Cell Signaling). Raw p-ERK values were normalised to vinculin (1:30; #4650; Cell Signaling). RAS activity was determined using the Ras activation Assay Biochem Kit (Cytoskeleton Inc. #BK008), according to manufacturer's instructions and quantified using an LiCor Odyssee CXL scanner and LiCor Image Studio software.

**Immunohistochemistry.** Immunohistochemistry was done on PFA-fixed and paraffin-embedded tissues. Organoids were fixed in 4% PFA for 30 min, while intestines were fixed overnight at room temperature. Subsequently, tissues were dehydrated in a graded ethanol series, followed by xylene. Tissues were paraffin-embedded, sectioned at 4 μm and mounted on Superfrost Plus slides (Thermo Fisher Scientific). Sections were deparaffinised, rehydrated, bleached for 10 min in 3% H_2O_2. Antigens were retrieved using 10 mM Na-citrate, pH 6 for 20 min at boiling temperature. The following antibodies were used: P-ERK (T202/Y204; #4370 CellSignal); P-MEK (S217/221; #9121 CellSignal); anti-RFP (1:200; #600-401-379 Rockland). ImmPRESS secondary antibody and NovaRED substrate kits (Vector Labs, Burlingame, CA, USA) were used for signal detection, according to manufacturer's protocols.

**Immunofluorescence and microscopy.** For immunofluorescence imaging, organoids were washed with PBS and fixed in-well with 4% PFA for 30 min at 37 °C. Fixation was stopped with PBS containing 100 nm Glycine. Cells were blocked and permeabilised with blocking buffer (PBS containing 1% BSA, 0.2% Triton X-100, 0.05% Tween-20) for at least 2.5 h at room temperature. Samples were incubated for 36 h at 4 °C with primary antibody against lysozyme (1:250; ab108508, Abcam) diluted in blocking buffer. After washing with IF-buffer (PBS containing 0.1% BSA, 0.2% Triton X-100, 0.05% Tween-20), samples were incubated for 24 h at 4 °C with secondary antibody Alexa Fluor 647 anti-rabbit (1:500, 4414, Molecular Probes) diluted in IF-buffer. Samples were counterstained for 5 min at room temperature using 0.5 μg/ml DAPI. After washing with IF-buffer, stained cultures were released from the Matrigel and collected in PBS. Samples were washed, resuspended in remaining PBS and mounted on slides using Vectashield Antifade Mounting Medium (H-1000, Vector).

Immunofluorescence and FIRE reporter images were taken with a Leica TSC SPE confocal microscope using an ACS 20× oil-immersion objective, solid-state lasers (405, 488, 532 and 635 nm) as sources of excitation and LAS X operating

software (Leica). Light microscopy images of cultures were taken with a Biozero microscope using a Plan Apo 4× NA 0.20 objective and Biozero observation and analyser software (Keyence).

For transmission electron microscopy, organoids were induced for 24 h, removed from Matrigel and fixed overnight at 4 °C in Karnovsky's fixative containing 2% PFA and 2.5% glutaraldehyde in PBS, washed three times in PBS and embedded in 1% LM-agrose. Agorose cubes (~1 mm³) containing several organoids were subsequently stained, using 0.5% $OsO_4$ in PBS (2.5 h at RT), 0.1% Tannic Acid in 100 mM Hepes pH 7.4 (1 h at RT) and 2% Uranyl acetate (1.5 h at RT), each time followed by three washing steps in PBS. Samples were dehydrated in a graded ethanol series, embedded in Spurr's resin, thin-sectioned (70 nm, Leica UC7) and post-contrasted as described[64]. Regions of ~100–150 µm² showing representative sections through organoids were imaged on a 120 kV Tecnai Spirit transmission electron microscope (FEI) equipped with a F416 CMOS camera (TVIPS). Micrographs were recorded automatically at a final magnification of 4400 × (2.49 nm pixel size at object scale) and −10 µm defocus using Leginon[65] and then stitched using TrakEM2[66].

**Flow cytometry and fluorescence-activated cell sorting**. Flow cytometry of anti-p-ERK-stained CRC cells resuspended in PBS/1% BSA cells was done using an Accuri cytometer (BD). Cells were gated for populations displaying different tdTomato fluorescence values (negative, low, medium and high), which correlates with transgene expression. For each population, the mean anti-p-ERK fluorescence values were determined and normalised to the tdTomato negative fraction of the corresponding FLUC control experiment.

For fluorescence-activated cell sorting of organoid cells, single-cell suspensions from induced organoids were prepared by digestion with TrypLE (Thermo Fisher Scientific) in the presence of 2 mM EDTA and 200 U/ml DNAse I. Digestion was monitored by visual inspection and stopped by crypt culture medium containing 0.2% BSA. Cell suspensions were filtered through 30 µm Celltrix filters and stained with an anti-CD44-antibody conjugated to Allophycocyanin (APC; clone IM7, BioLegend) and the Green or Near-IR Live/Dead Fixable Dead Cell Stain Kits (Life Tech) for subsequent exclusion of dead cells. Single cells were sorted into the 96-well plates of the Precise WTA Kit with predispensed library chemistry using a BD FACSAriaII SORP (BD) and a gating strategy as displayed in Supplementary Fig. 3. Cells were sorted into quadrants of plates to minimise batch effects between plates. For later analysis of CD44 positivity of the subsets, sorts were performed as index sorts.

**RNA sequencing and bioinformatic analyses**. For single-cell RNA sequencing, the Precise WTA Kit (BD) was used, according to the manufacturers' instructions. Sequences were produced using NextSeq and/or HiSeq chemistry (Illumina). Cluster generation on NextSeq 500 followed the instructions of the manufacturer, at a final loading concentration of 2 pM on a high-output-flowcell. 1% PhiX was added as quality control, at least 40 million paired reads per pool were gained during a Paired-End-75 run. Library-pools running on the HiSeq4000-system were prepared according to Illumina recommendations, loaded with 200 pM concentration and sequenced during a Paired-End-75 run. Again, 1% PhiX was added as quality control, and at least 40 million read-pairs per pool were targeted.

Single-cell RNA-sequencing data were pre-processed using the BD Precise Whole Transcriptome Assay Analysis Pipeline v2.0[67]. Quality control was performed using scater[68]. Read counts were normalised using the trimmed mean of median values (TMM) approach provided with edgeR[69]. Normalised read counts were used for k-means clustering and t-SNE visualisation. Differentiation trajectories in t-SNE plots were determined using slingshot[70], with intestinal stem cell cluster 1 as predefined origin. Differentially expressed genes were called on log-transformed raw counts using a hurdle model provided with R package MAST[71]. Top-10 signature genes per cluster were identified by comparing average gene expressions within cluster to average gene expressions across all other clusters. For bulk cell RNA sequencing, organoids were induced for 24 h with 2 µg/ml doxycycline in CCM-REN medium and subsequently dissociated using TrypLE (Thermo Fisher Scientific). RNA-seq reads were aligned to the mouse genome GRCm38 using STAR aligner with GENCODE annotation vM11. Differentially expressed genes were called using DESeq2.

**Mathematical modelling**. We quantified and locally adjusted the signalling networks from the KRAS-mutant perturbation data using an adjusted version of Modular Response Analysis as implemented in the R package STASNet, version 1.0.2[72] (https://github.com/molsysbio/STASNet/tree/STASNet1.0.2) as follows: As input data, we derived representative mean and standard error of the mean values from the single-cell CyTOF data (trimming the lower and upper 5% of signals) and a literature-informed prior network. As the apparent response pattern across clusters was similar, it was decided to use a joint modelling approach, i.e. we first quantified the response coefficients for all six clusters by a single set of coefficients using a combination of latin hypercube sampling with subsequent Levenberg-Marquardt fitting ($n = 4 \times 10^4$) and then iteratively derived and quantified the significantly different signalling coefficients between clusters using a likelihood ratio test. Afterwards we searched for biologically justifiable link extensions lacking

in the network to better describe the data. The whole procedure was repeated until no further justifiable link additions could be found, followed by a removal round of statistically insignificant links.

**Reporting summary**. Further information on research design is available in the Nature Research Reporting Summary linked to this article.

## Data availability

scRNA-seq and bulk RNA-seq data are available from GEO repository under accession numbers GSE115242 and GSE115234, respectively. Data underlying Figs. 1, 6, Supplementary Fig. 7 (CyTOF data) and Fig. 2b, d, e, and Supplementary Figs. 1c, 6, 8, 9 are provided as source data files. Modelling information is available as a source data file.

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

## Acknowledgements

The authors gratefully acknowledge excellent technical assistance by Anja Sieber (IRI Life Sciences and Charité Universitätsmedizin Berlin), Gaby Bläss and Sonja Banko (MPIMG, Berlin) for RAS activity assays, mouse genotyping and mouse care, respectively. The authors also gratefully acknowledge the help of lab students Ekaterina Eroshok and Maximilian Anders (Molecular Medicine Masters programme, Charité Universitätsmedizin Berlin) with immunohistochemistry in the early phase of this project. We received the FIRE plasmid as a kind gift from John Albeck, UC Davis. The work was in part funded by Deutsche Forschungsgemeinschaft (MO2783/2-1 to M.M.), Berlin School of Integrative Oncology (to N.B. and M.M.), the German Cancer Consortium DKTK (to N.B. and M.M.), the Federal Ministry of Education and Research BMBF (StemNet 01EK1604B to N.B.; ColoSys 031L0081 to C.S. and N.B.) and the Berlin Institute of Health (to N.B., C.S. and M.M.).

## Author contributions

R.B., T.S., M.L., P.R., C.G.-T., S.S., D.K., N.M., B.F. and I.A.E.-S. conducted, analysed and interpreted experiments; F.U., T.S. and B.K. performed bioinformatic analyses; M.M., N.B., C.S., P.R., T.M. and B.G.H. conceived, designed, interpreted experiments and/or supervised parts of the study; M.M., N.B. and B.K. wrote the paper.

## Additional information

**Competing interests:** The authors declare no competing interests.

