## [Peer Review File · Nature Communications]

Reviewers' comments:

Reviewer #1 (Remarks to the Author):

Brandt et al., using a combination of transgenic models and single cell seq describe cell-type specific activation of MAPK kinase signalling downstream of mutant Kras or Braf in intestinal epithelium. The main strength of the study is the use of scRNAseq approach on identifying cell specific MAPK activation signatures in mouse intestinal epithelium. However, the weaknesses of the experimental system employed and lack of in vivo relevance of the identified cell signatures make it difficult to be useful for the larger CRC community.

Major Comments to Brandt et al.,

1. The authors choose to study BrafV600E mutation and the KrasG12V mutation in MAPK activation employing a transgenic RosaCre recombinase driven activation of these mutations. This a major issue with the study, why use RosaCre for activation of these oncogenes? There are genetically engineered mouse models of both of these genes (LSL-KrasG12V and BrafV600E) available in their endogenous locus. The authors should study these genetic mutations in their endogenous locus; this will make the study relevant in the context of present knowledge of the role of these mutations in intestinal cancers.
2. Though Kras is a frequent mutation in CRC, it is well known that KrasG12D variant is the most frequently occurring mutation of Kras in CRC and many other cancer types. Moreover, it is also known that the G12D mutant version of Kras is also biochemically and functionally more active than KrasG12V. Using the 12V variant, makes authors draw a conclusion that Kras being less-able to activate ERK in similar cell types as BrafV600E. The authors need to use the KrasG12D mutant version to do the scRNASeq and corroborate this further.
3. Authors use FIRE reporter to assess ERK activation status that is based on Fra-1 activity. However Fra-1 is not the only downstream target of ERK activation. Authors should show that CRISPR of Fra-1 leads to the rescue in ERK activity they observe in Braf and Kras cells.
4. Finally, following the above mentioned lower activity of the KrasG12V, authors use B-catenin together with Kras12V to regulate ERK phosphorylation in CRC. In contrary to what the authors suggest as conventional CRC progression, Kras mutation doesn't necessarily have to follow Wnt activation and there is emerging data suggesting that Kras mutant CRC can acquire Wnt activation at later stages and also shown in (See Bennecke et al, Cancer cell. 2010).
5. There is a lack of application of the data to human relevance. Authors should first validate and corroborate the signatures derived from scRNAseq and to identify human CRC relevant signatures in the Braf and Kras mutant tumours.

Minor Comments,

1. The BrafV600E and the KrasG12V mutant mice used to derive organoids in the study, where they homozygous or heterozygous for the mutations?

Reviewer #2 (Remarks to the Author):

KRASG12V and BRAFV600E are major oncogenes that drive ~40% and ~10% of all colorectal cancers respectively. It is generally thought that these two oncogenes transmit their signals primarily via the ERK signaling cascade that further induces enhanced proliferation. In the present study, the authors explored cell type-dependent KRASG12V and BRAFV600E-mediated oncogenic signaling in organoids of mouse intestinal epithelium. Surprisingly, they found that KRASG12V

induces ERK activation as well as proliferation only of a subset of the cells, while BRAFV600E induces ERK activation in all cell types in the organoid. Further experiments suggest that the Wnt/ β -Catenin signaling may maintain EGF and KRASG12V-induced proliferation only in adenomatous cells explaining why activation of Wnt signaling precedes KRAS mutations in the classical CRC progression pathway.

Overall this is an interesting study that highlights key differences between ERK activation upon KRASG12V and BRAFV600E expression in colorectal cancer. Some of the effects can be regulated/mediated, at least in part, by the Wnt/ β -Catenin pathway, thus having a fundamental relevance for the study of colorectal cancer therapy. The system of KRASG12V and BRAFV600E expressing organoids established by the authors is very impressive, the manuscript is well written, and the results are believable. However, at this stage the manuscript seems to be premature as it lacks information on the mechanistic as well as clinical aspects of the differences. These, as well as a few other comments are listed below.

1. The authors established an important experimental system of inducible transgenic expression of the two oncogenes: KRASG12V and BRAFV600E in intestinal organoid. The paper describes the establishment of the system and nicely demonstrates the very interesting observation of lack of ERK activation in some of the intestinal cell types. However, the molecular mechanisms that allow this phenomenon are not properly deciphered, as it is probably not fully related to the Wnt/ β -Catenin signaling. No other mechanisms have been shown. The mechanism that allow the differences between the cell types should be investigated in order to make the paper suitable for publication in Nature Communications. It is well known that the activation of ERK by KRASG12V varies in intensity between cell lines and growing conditions. This is usually attributed to overexpression of phosphatases, or other negative feedback regulators. I believe that the sequencing information generated by the authors may provide a clue as to what are the components that are involved here.

2. The authors suggest that ERK is not activated in all cells types even after EGF stimulation, not just KRASG12V expression. What is the reason for this effect. Do these cells express EGFR? Can EGF activate any other signaling cascade in these cells (e.g. AKT)? Is overexpression of phosphatases involved in this effect?

3. In figure 1 and 3, it would be important to study ERK phosphorylation using western blots, not just capillary protein analysis. This would provide more sensitive information on small activation and whether both ERK1 and ERK2 are equally affected. In addition, staining of the organoids with anti ERK1 and ERK2 and providing bigger magnification of the pERK staining would provide information on the subcellular localization of the ERKs, which might be involve in their inability to be activated as well.

4. There is no mention of the correlation between the results here and results obtained from patients' biopsies.

5. In Fig. 3B the staining is not clear, and I am not sure what does the Asterix shows. A better and clearer picture should be provided. In addition, it is suggested to include staining of KRASG12V and BRAFV600E organoids in the same magnification.

6. In Fig. 3D, staining of ERK, pERK and Ki67 of both KRASG12V and BRAFV600E with a slightly bigger magnification should be provided. It will be nice to include staining of pAKT as well.

7. The authors used Rho kinase inhibitor Y27632 (10 μ M) to prevent anoikis. Rho kinase is not the only mediator of anoikis, and it is possible that it is affected by other mechanisms tht are induced by the oncogenes examined.

8. It is recommended to examine the role of Raf and MEK inhibitors on the structure and growth of the organoids.

9. The terminology used is sometime problematic:

a) The author use the terms MAPK and ERK interchangeably, without any logic behind it. In particular the phrase "MAPK driven ERK targets" (page 10 line 18) is strange. The term MAPK is usually used for the family of MAPKs including ERK, JNK p38 and ERK5, while ERK is kept for ERK1 and ERK2 that are the component of the unique ERK cascade. It is suggested to use ERK or ERK cascade throughout the whole article.

b) The term "functional heterogeneity" is not well-explained and might be unjustified.

- c) The sentence in page 10 lines 16-17 might be problematic as ERK can be activated in these cells using the expression of BRAFV600E.
- d) Page 9, line 10, should be Fig. 3B.
- e) Page 13 line 24, is it survival or proliferation?
- f) A suitable reference should be given to line no 11 of the discussion section stating as ERK levels are generally higher in cancer cells adjacent to stromal cells at the invasive front, and lower in more central areas.

The authors would like to thank both reviewers for taking their time to assess our manuscript. As outlined below, we almost completely agree with the assessment by reviewer #2. However, we mostly disagree with the rather negative views of reviewer #1 that, in our opinion, were compromised by a couple of technical misunderstandings. We hope that that our answers address these points in sufficient depth for reviewer #1, as some of these technical points cannot be clarified in the form of new experiments.

Reviewer #1

Brandt et al., using a combination of transgenic models and single cell seq describe cell-type specific activation of MAPK kinase signalling downstream of mutant Kras or Braf in intestinal epithelium. The main strength of the study is the use of scRNAseq approach on identifying cell specific MAPK activation signatures in mouse intestinal epithelium.

However, the weaknesses of the experimental system employed and lack of in vivo relevance of the identified cell signatures make it difficult to be useful for the larger CRC community.

We think that the perceived weakness of our experimental approach is in fact a strength, as our doxycycline-driven transgene system that expresses oncogenes along with a fluorescent marker allowed us to perform single cell studies in a way that cannot be accommodated easily using cre-recombinase LSL models. We have outlined this below in response to major point 1.

We have in this extensively revised version of our manuscript expanded our single cell experiments that were described above as the main strength of the study: we now include both single cell RNA seq and CyTOF experiments, and furthermore, we have also included single cell data from patient-derived organoids. Finally, we have used quantitative modelling and new RNA-seq analyses to provide a mechanistical basis for differential KRAS to ERK signaling in cell types of the intestinal epithelium. We hope that reviewer #1 finds the revised manuscript in its present form to be useful for the larger CRC community.

Major Comments to Brandt et al.,

1. The authors choose to study BrafV600E mutation and the KrasG12V mutation in MAPK activation employing a transgenic RosaCre recombinase driven activation of these mutations. This a major issue with the study, why use RosaCre for activation of these oncogenes? There are genetically engineered mouse models of both of these genes (LSL-KrasG12V and BrafV600E) available in their endogenous locus. The authors should study these genetic mutations in their endogenous locus; this will make the study relevant in the context of present knowledge of the role of these mutations in intestinal cancers.

The reviewer describes our model as “RosaCre recombinase driven”, and states this as a major concern. However, we employ a doxycycline-driven inducible system, not a cre-recombinase system. This system is different from cre-recombinase/lox-stop-lox systems and it has its own set of strengths, such as the integration into a single locus, and the fluorescent marker coupled directly (in the same transcript) to the oncogene transgene, allowing us to directly compare transgene induction and reporter output on a cell-to-cell basis, as in Figure 3 of the initial manuscript (now Figure 4 of the revised manuscript). This approach would be very complicated to replicate with LSL-oncogene models, as suggested by the reviewer.

To avoid misunderstandings about the nature of the transgene system for future readers of our paper, we have now included a graphical representation of the transgene system in Figure 2a, in addition to the descriptions and the references in the results and methods parts.

Regarding the argument of expression strength in transgene versus endogenous locus, we would like to point out that oncogenes, including KRAS and BRAF, are frequently overexpressed in cancer. Thus, we think that transgenic expression does not invalidate our results. Indeed, research we published previously using the BRAF model (1) was recently confirmed using an LSL-BRAF^{V600E} model (2), clearly implying that research on alternative mouse models can complement each other.

2. Though Kras is a frequent mutation in CRC, it is well known that KrasG12D variant is the most frequently occurring mutation of Kras in CRC and many other cancer types. Moreover, it is also known that the G12D mutant version of Kras is also biochemically and functionally more active than KrasG12V. Using the 12V variant, makes authors draw a conclusion that Kras being less-able to activate ERK in similar cell types as BrafV600E. The authors need to use the KrasG12D mutant version to do the scRNASeq and corroborate this further.

We agree that KRAS G12D is slightly more prevalent compared to G12V in human colon cancer, however not by a large margin: in the hallmark TCGA study on CRC published in Nature (3), there are 28 KRAS G12D versus 22 G12V mutations in the cohort (and 10 G13D, 6 G12C as the next most prevalent mutations). Similar frequencies were also found in a more recent study published in Cancer Cell on 1099 CRC patients (4), where the authors found 101 G12V vs. 137 G12D mutations. We believe that all frequent RAS mutations are relevant targets of research.

Indeed, cre-recombinase LSL models of both, KRAS G12D and G12V have been generated (such as these from the Jacks and Sansom labs, respectively (5–7)), and both have contributed to our understanding of mutant KRAS. However, to our knowledge, KRAS G12D and G12V mouse models have never been analysed side by side in the same genetic background and in the same study, making any judgements based on mouse

models alone incomplete. Importantly, both KRAS mutations were shown to be competent oncogenes in the mouse intestine, complementing the data from human CRC.

The reviewers' assessment that G12D is biochemically and functionally more active than G12V is not referenced and in our opinion misguided. Nor would this argument - even if it was true - make research into the 2nd most frequent KRAS mutation in CRC irrelevant. To our knowledge, and after extensive literature studies, we find that there are only few papers in the field correlating different KRAS mutations at the same codon side-by-side, and there is no consensus in the field as to their differential impact, biochemically or clinically. Codon 12 and codon 13 mutations appear to differ slightly in biochemical properties (such as residual GTPase activity) as well as in clinical outcome (8, 9). Contrary to the reviewers claim that G12D is more active than G12V, early biochemical analysis even suggested that G12V RAS proteins have less residual GTPase activity compared to G12D mutants, making them more potent oncoproteins (10).

3. Authors use FIRE reporter to assess ERK activation status that is based on Fra-1 activity. However Fra-1 is not the only downstream target of ERK activation. Authors should show that CRISPR of Fra-1 leads to the rescue in ERK activity they observe in Braf and Kras cells.

The reviewer seems to have a wrong understanding of the reporter, as it is not based on Fra-1 activity. The reporter is a constitutively expressed fluorescent protein that is directly stabilized by ERK post-translationally at consensus phosphorylation sites (as present in Fra-1, hence the name) as an established tool for reporting time-integrated nuclear ERK activity(11), while making no assumptions about downstream target genes in individual cells. Therefore, inactivation of Fra-1 has no relevance as a control to validate the reporter, and indeed, we would expect reporter activity to be unchanged after deletion of Fra-1.

We have included a graphical representation of the pathway and the reporter in Figure 4a, to aid readers in interpretation of our data.

4. Finally, following the above mentioned lower activity of the KrasG12V, authors use B-catenin together with Kras12V to regulate ERK phosphorylation in CRC. In contrary to what the authors suggest as conventional CRC progression, Kras mutation doesn't necessarily have to follow Wnt activation and there is emerging data suggesting that Kras mutant CRC can acquire Wnt activation at later stages and also shown in (See Bennecke et al, Cancer cell. 2010).

Indeed, we outline in our introduction the widely accepted view that both BRAF and KRAS can be alternative initiators of the serrated progression pathway, and the study cited by the reviewer provides insight into the

mechanisms of KRAS as an initiator in the serrated pathway. We fully appreciate the work presented there, but fail to understand why and how data presented in Bennecke et al. (12) could be a potential argument against our study. Indeed, there is a well-known and strong disequilibrium between KRAS and BRAF mutations in the conventional versus serrated progression pathways with KRAS mutations mostly (but not exclusively) occurring in the former after APC inactivation (13). This is what we discuss, and it is in no way contradictory with the findings cited by the reviewer.

5. There is a lack of application of the data to human relevance. Authors should first validate and corroborate the signatures derived from scRNAseq and to identify human CRC relevant signatures in the Braf and Kras mutant tumours.

We agree that the focus of the previous version of our manuscript was on oncogenic signal transduction with unclear translational impact. We now addressed this by analysis of patient-derived organoids. The new data on human organoids is presented in the new Fig. 1 of the manuscript. It is of note that our analysis of human cultures is limited to KRAS/BRAF-wildtype and KRAS-mutant organoids, as we have no BRAF-mutated patient-derived organoid line available. The cancers are rare to start with (10% of all CRCs, and even less MSS cancers), and organoids could not be established from the two BRAF-mutant patients that were available to us.

We hope that reviewer #1 will also appreciate our modeling studies in mouse organoids that lead to a better mechanistic understanding of cell type-specific KRAS to ERK signal transduction that is now provided in the new Figures 6 and 7 of the manuscript.

Minor Comments,

1. The BrafV600E and the KrasG12V mutant mice used to derive organoids in the study, were they homozygous or heterozygous for the mutations?

Our transgene system is located in the Rosa26 locus, as already discussed above, major point one. All organoids/mice used are heterozygous for the transgene system, as the mice were derived from crossings of homozygous mice to C57Bl/6J wildtype animals.

Reviewer #2

KRASG12V and BRAFV600E are major oncogenes that drive ~40% and ~10% of all colorectal cancers respectively. It is generally thought that these two oncogenes transmit their signals primarily via the ERK signaling cascade that further induces enhanced proliferation. In the present study, the authors explored

cell type-dependent KRASG12V and BRAFV600E-mediated oncogenic signaling in organoids of mouse intestinal epithelium. Surprisingly, they found that KRASG12V induces ERK activation as well as proliferation only of a subset of the cells, while BRAFV600E induces ERK activation in all cell types in the organoid. Further experiments suggest that the Wnt/ β -Catenin signaling may maintain EGF and KRASG12V-induced proliferation only in adenomatous cells explaining why activation of Wnt signaling precedes KRAS mutations in the classical CRC progression pathway.

Overall this is an interesting study that highlights key differences between ERK activation upon KRASG12V and BRAFV600E expression in colorectal cancer. Some of the effects can be regulated/mediated, at least in part, by the Wnt/ β -Catenin pathway, thus having a fundamental relevance for the study of colorectal cancer therapy. The system of KRASG12V and BRAFV600E expressing organoids established by the authors is very impressive, the manuscript is well written, and the results are believable. However, at this stage the manuscript seems to be premature as it lacks information on the mechanistic as well as clinical aspects of the differences.

We almost completely agree with the assessment of the reviewer, including the perceived weaknesses with regard to the mechanistic as well as translational aspects. We now addressed these limitations of our original manuscript as stated below.

1. The authors established an important experimental system of inducible transgenic expression of the two oncogenes: KRASG12V and BRAFV600E in intestinal organoid. The paper describes the establishment of the system and nicely demonstrates the very interesting observation of lack of ERK activation in some of the intestinal cell types. However, the molecular mechanisms that allow this phenomenon are not properly deciphered, as it is probably not fully related to the Wnt/ β -Catenin signaling. No other mechanisms have been shown. The mechanism that allow the differences between the cell types should be investigated in order to make the paper suitable for publication in Nature Communications. It is well known that the activation of ERK by KRASG12V varies in intensity between cell lines and growing conditions. This is usually attributed to overexpression of phosphatases, or other negative feedback regulators. I believe that the sequencing information generated by the authors may provide a clue as to what are the components that are involved here.

We thank the reviewer for the important suggestion of consulting our RNAseq data for overexpression of phosphatases or other negative regulators. When assessing the data in our original submission, we found that the clues for a mechanistical basis of the cell type-specific ERK activation could not be derived from our single cell sequencing data, as these data were too sparse with respect to transcriptome coverage.

We therefore carried out two lines of research. One of the approaches was to use RNA seq data from bulk sorted FIRE-high versus FIRE-low cells (harvested in parallel to the single cells used in the single cell study, Fig. 5). The second approach was to assess the underlying signaling networks directly on a cell-to-cell basis using cytometry by mass spectroscopy (CyTOF)(14). For this, we used a dedicated CyTOF antibody panel for analysis of CRC cells that we have generated and tested in the last two years (a separate manuscript describing the panel in-depth is currently in preparation). The CyTOF panel covers antibodies binding cell-type specific and surface markers, as well as antibodies directed against key signal transducer (phospho-)proteins.

As described in the revised manuscript, we now generated a perturbation data set using the mouse KRASG12V- and FLUC control transgenic organoids. We used the surface markers in the CyTOF panel to assign cells into six distinct clusters. We next used the signal transducer information to quantitatively model the underlying signaling networks in a cluster specific manner, using Modular Response Analysis (15).

Most importantly, we compared the two clusters with the highest p-ERK levels after KRAS-induction with the two clusters that were not at all KRAS-responsive and had lowest p-ERK level. We found that the underlying signaling networks were very similar, but differed in two crucial points: KRAS-to-ERK-responsive cells had higher levels of MEK-to-ERK feed forward signaling, but lower levels of feedback from ERK to MEK (the latter is indicative of any feedback from ERK to upstream pathway components. Our model cannot further define the molecular nature or wiring of the feedback).

Finally, we consulted the bulk RNA-seq data from FIRE-high versus FIRE-low sorted cells, and identified three DUSP family phosphatases with differential expression. As these are ERK-specific, we suggest that these could be implicated in the differential regulation of MEK-to-ERK signaling.

In hindsight, the reviewer's comment was surprisingly spot-on, regarding signaling feedback and phosphatases.

The new data is shown in our new Figures 6 and 7, and described in the results section starting p.8, line 28)

2. The authors suggest that ERK is not activated in all cells types even after EGF stimulation, not just KRASG12V expression. What is the reason for this effect. Do these cells express EGFR? Can EGF activate any other signaling cascade in these cells (e.g. AKT)? Is overexpression of phosphatases involved in this effect?

As described above, we can now at least provide partial answers to these questions. For one, we have the bulk RNAseq data of FIRE-low versus FIRE-high cells. We assessed transcription of all genes encoding MAPK

pathway components, among these the EGFR family genes. We found no differential expression (which of course does not mean that there is no difference in protein levels or activities).

Furthermore, our CyTOF panel assessed differential activities in signaling pathways interacting with RAS-MEK-ERK across cells across differentiation states (as represented by the six clusters in Fig. 6C). We saw that signaling levels were different between the cell types, however quantitative modelling only identified few differences in network interactions. Our antibody panel includes an anti p-AKT(S473) antibody that was positively evaluated during panel testing using IGF-stimulated HCT116 CRC cells. In the new experiments with mouse organoids in the manuscript, the antibody showed only weak signals across all clusters, precluding a detailed analysis. As the AKT downstream phosphorylation sites p-4EBP1 (T37/46) and pS6 (S235/236) were in panel, we could assess signal transduction via AKT, and our quantitative model suggests no gross differences between clusters.

These data can be found in the new Fig. 7, and is described in the results section starting page 9, line 28.

3. In figure 1 and 3, it would be important to study ERK phosphorylation using western blots, not just capillary protein analysis. This would provide more sensitive information on small activation and whether both ERK1 and ERK2 are equally affected. In addition, staining of the organoids with anti ERK1 and ERK2 and providing bigger magnification of the pERK staining would provide information on the subcellular localization of the ERKs, which might be involve in their inability to be activated as well.

Our work with organoids cultures (as opposed to cell lines) complicates analysis by Western blotting, as the cultures are smaller and do not yield as much material as cell cultures. Additionally, we think that CyTOF is a superior tool compared to Westerns within the context of our manuscript, since these data provides cellular resolution, and thus shows us pERK differences in subgroups/clusters of cells. But of course our experimental approach comes with its own limitations: the reviewer is right that we do not provide any information to distinguish between ERK1 and ERK2 activities, as our antibody detects both phosphorylated proteins. Whether the two isoforms of ERK have functional differences or are functionally redundant is an ongoing debate and there are a number studies that tried to address this (reviewed in ref. 16 below). Given our focus on cell type specificity, we cannot address this important separate question here.

Regarding the question of subcellular localization, our IHC data is not good enough to provide unequivocal information. Indeed, pERK levels are known to oscillate (11), and IHC only gives a snapshot of ERK activity from cell to cell, making it difficult to extract meaningful information from magnifications of few cells. Nevertheless, the FIRE data (Fig. 4 of the revised manuscript) measures nuclear ERK activity only. When we compare the total pERK data by CyTOF with FIRE activity, we see a very good agreement all cell types, except for the presumptive undifferentiated crypt cells in cluster 1 (cluster definition as in Fig. 6C): These show

high FIRE activity but only intermediate pERK levels in CyTOF. We think that these findings could be due to further levels of ERK regulation, such as subcellular localization, as discussed in the revised manuscript (Discussion, page 13, line 15 and following).

4. There is no mention of the correlation between the results here and results obtained from patients' biopsies.

For the revised manuscript, we looked into patient-derived organoid cultures (17) instead of biopsies, as suggested by the reviewer. Using CyTOF allowed us now to correlate pMEK and pERK levels with cell differentiation markers in organoids. We show differentiation trajectories of patient-derived CRC lines from EphB2-high (=undifferentiated) to cleaved-Caspase-high (=apoptotic at end of lifespan). pMEK and pERK levels correlate with EphB2, assigning their activities to undifferentiated or crypt-like cells, irrespective of their KRAS mutational status. This shows that our findings in transgenic mouse organoids are relevant also in CRC.

The new results are shown in Fig. 1B, and are described in the results section of the revised manuscript, page 5.

5. In Fig. 3B the staining is not clear, and I am not sure what does the Asterix shows. A better and clearer picture should be provided. In addition, it is suggested to include staining of KRASG12V and BRAFV600E organoids in the same magnification.

We show now the FIRE fluorescent panel on its own in Figure 4B (former Fig 3B), in order to provide a clearer picture. Crypt and villus areas are now marked c and v, respectively. Fig. 4C (formerly 3C) shows KRASG12V and BRAFV600E organoids in the same magnification (see size bar in first panel).

Asterisks in Fig. 4B and 4C show isolated FIRE-positive villus cells, as also stated in the legend to Fig. 4, page 15 top. Across our experiments, we repeatedly found this unexpected subpopulation of ERK/FIRE-positive villus cells: as seen in Fig. 5D and Supplementary Fig. 5, these appear to be late stage enterocytes, as judged by gene expression. We believe that these cells represent cluster 5 in our new CyTOF data (Fig. 6 C-G; cluster 5 cells are identified as a population of p-ERK-high cells with low levels of crypt markers, many of these are c-Casp3-positive). It is of note that cells with similar characteristics have been identified before, for instance in the paper Simmons et al. from Ken Lau's lab (18).

6. In Fig. 3D, staining of ERK, pERK and Ki67 of both KRASG12V and BRAFV600E with a slightly bigger magnification should be provided. It will be nice to include staining of pAKT as well.

The purpose of this figure (now Fig 4D of the revised manuscript) is to provide information on organoid domains (such as, crypt versus villus domain), and not subcellular information in single cells (see also point 3, above). It also serves as an independent information for the FIRE reporter assays. We believe that higher magnifications covering only partial organoids would make it harder to distinguish domains. Regarding the pAKT staining, we agree that this information would be valuable, however pAKT staining in IHC are artifact-prone in our experience. We now used AKT and downstream targets in our CyTOF analysis, see above (point 2).

7. The authors used Rho kinase inhibitor Y27632 (10 μ M) to prevent anoikis. Rho kinase is not the only mediator of anoikis, and it is possible that it is affected by other mechanisms tht are induced by the oncogenes examined.

Y27632 was not used for continuous culture, but only for thawing and viral transfections of organoids, and during passaging of spheroids induced by stabilized beta-catenin, as stated in the methods section. For these steps, it is a standard (and effective) additive to culture medium to prevent anoikis, as given in the respective references. We do not draw any further conclusions about Y27632s role and our experimental results are not affected by it, as it is not used in our usual culture medium. We have re-written the respective part of the methods section to avoid misunderstandings.

8. It is recommended to examine the role of Raf and MEK inhibitors on the structure and growth of the organoids.

In the new network analysis, we employed GSK3 β , MEK, p38 (and PI3K, later removed from the dataset, due to lack of discernable effects) inhibitors, to cover a maximum of perturbations for network reconstruction. However, we only assess organoid growth and composition (as changes of cell fractions between clusters 1-6) for KRASG12V induction and GSK3 β inhibition (that is, activation of RAS and Wnt/beta-Catenin signaling). The other inhibitors are used only short-term and thus cannot affect growth. We think that our focus of signal transduction is justified, and further inhibitions within the RAS-ERK pathway would add little information to the manuscript with its present focus.

9. The terminology used is sometime problematic:

a) The author use the terms MAPK and ERK interchangeably, without any logic behind it. In particular the phrase "MAPK driven ERK targets" (page 10 line 18) is strange. The term MAPK is usually used for the family

of MAPKs including ERK, JNK p38 and ERK5, while ERK is kept for ERK1 and ERK2 that are the component of the unique ERK cascade. It is suggested to use ERK or ERK cascade throughout the whole article.

We agree and have modified the manuscript accordingly.

b) The term “functional heterogeneity” is not well-explained and might be unjustified.

We agree and have removed the term.

c) The sentence in page 10 lines 16-17 might be problematic as ERK can be activated in these cells using the expression of BRAFV600E.

d) Page 9, line 10, should be Fig. 3B.

e) Page 13 line 24, is it survival or proliferation?

These sentences have vanished during the extensive re-writing of the manuscript during the revision.

f) A suitable reference should be given to line no 11 of the discussion section stating as ERK levels are generally higher in cancer cells adjacent to stromal cells at the invasive front, and lower in more central areas.

The reference is Blaj et al., *Cancer Res.* 2017 (19), and it is now referenced at multiple positions in the manuscript (including the position mentioned by the reviewer, now page 12, line 17).

References

1. Riemer P et al. Transgenic expression of oncogenic BRAF induces loss of stem cells in the mouse intestine, which is antagonized by β -catenin activity. [Internet]. *Oncogene* 2015;34(24):3164–3175.
2. Tong K et al. Degree of Tissue Differentiation Dictates Susceptibility to BRAF-Driven Colorectal Cancer. [Internet]. *Cell Rep.* 2017;21(13):3833–3845.
3. The Cancer Genome Atlas Network. Comprehensive molecular characterization of human colon and rectal cancer [Internet]. *Nature* 2012;487(7407):330–337.
4. Yaeger R et al. Clinical Sequencing Defines the Genomic Landscape of Metastatic Colorectal Cancer. *Cancer Cell* [published online ahead of print: 2018]; doi:10.1016/j.ccell.2017.12.004
5. Tuveson DA et al. Endogenous oncogenic K-rasG12D stimulates proliferation and widespread neoplastic and developmental defects. *Cancer Cell* [published online ahead of print: 2004]; doi:10.1016/S1535-6108(04)00085-6
6. Haigis KM et al. Differential effects of oncogenic K-Ras and N-Ras on proliferation, differentiation and tumor

- progression in the colon. [Internet]. *Nat. Genet.* 2008;40(5):600–608.
7. Sansom OJ et al. Loss of Apc allows phenotypic manifestation of the transforming properties of an endogenous K-ras oncogene in vivo. [Internet]. *Proc. Natl. Acad. Sci. U. S. A.* 2006;103(38):14122–14127.
 8. De Roock W et al. Association of KRAS p.G13D mutation with outcome in patients with chemotherapy-refractory metastatic colorectal cancer treated with cetuximab. *JAMA - J. Am. Med. Assoc.* 2010;304(16):1812–1820.
 9. Guerrero S et al. K-ras codon 12 mutation induces higher level of resistance to apoptosis and predisposition to anchorage-independent growth than codon 13 mutation or proto-oncogene overexpression. *Cancer Res.* 2000;60(23):6750–6756.
 10. Seeburg PH, Colby WW, Capon DJ, Goeddel D V., Levinson AD. Biological properties of human c-Ha-ras1 genes mutated at codon 12. *Nature* 1984;312(5989):71–75.
 11. Albeck JG, Mills GB, Brugge JS. Frequency-Modulated Pulses of ERK Activity Transmit Quantitative Proliferation Signals. *Mol. Cell* 2013;49(2):249–261.
 12. Bennecke M et al. Ink4a/Arf and oncogene-induced senescence prevent tumor progression during alternative colorectal tumorigenesis. [Internet]. *Cancer Cell* 2010;18(2):135–146.
 13. Morkel M, Riemer P, Sers C, Bläker H, Sers C. Similar but different: distinct roles for KRAS and BRAF oncogenes in colorectal cancer development and therapy resistance [Internet]. *Oncotarget* 2015;6(25):20785–20800.
 14. Bodenmiller B et al. Multiplexed mass cytometry profiling of cellular states perturbed by small-molecule regulators. *Nat. Biotechnol.* 2012;30(9):858–867.
 15. Kholodenko BN et al. Untangling the wires: A strategy to trace functional interactions in signaling and gene networks. *Proc. Natl. Acad. Sci.* [published online ahead of print: 2002]; doi:10.1073/pnas.192442699
 16. Buscà R, Pouyssegur J, Lenormand P. ERK1 and ERK2 Map Kinases: Specific Roles or Functional Redundancy?. *Front. Cell Dev. Biol.* [published online ahead of print: 2016]; doi:10.3389/fcell.2016.00053
 17. Schütte M et al. Molecular dissection of colorectal cancer in pre-clinical models identifies biomarkers predicting sensitivity to EGFR inhibitors [Internet]. *Nat. Commun.* 2017;8:14262.
 18. Simmons AJ et al. Cytometry-based single-cell analysis of intact epithelial signaling reveals MAPK activation divergent from TNF- α -induced apoptosis in vivo [Internet]. *Mol. Syst. Biol.* 2015;11(10):835–835.
 19. Blaj C et al. Oncogenic effects of high MAPK activity in colorectal cancer mark progenitor cells and persist irrespective of RAS mutations. *Cancer Res.* 2017;77(7):1763–1774.

Reviewers' comments:

Reviewer #1 (Remarks to the Author):

The authors have improved the manuscript in different sections; as suggested the authors have added data from patient-derived organoids which largely improves the manuscript. The clarification of the experimental systems used makes the manuscript easier to follow and puts the data in the context of other available scSeq datasets.

1. scSeq data from the patient-derived organoids complements the findings on mouse and improves the manuscript.

2. With regards to studying the KrasG12V mutant compared to G12D, though it is not the focus of the current study. I have to differ from the authors claim that these two mutants are similar. Though there hasn't been a head-to-head comparison in mouse models there is plenty of literature on characterizing the different Kras point mutations both biochemically and using isogenic cell lines. A few of these studies from the Westover and Prior Labs (Hunter et al., 2015; Hammond et al., 2015) are examples showing, that in-terms of biochemistry and proteome and phosphoproteome of these mutants is very different and that G12V has low activity than G12D and C. However, this does not make work from the authors on the G12V mutant irrelevant, it is important to acknowledge and be aware of work by others in understanding these mutations.

Minor comment:

There are a few typos during the rewrite.

page 5, line 25

page 7, line 21

<https://pubs.acs.org/doi/pdfplus/10.1021/pr501191a>

<http://mcr.aacrjournals.org/content/13/9/1325.long>

Reviewer #2 (Remarks to the Author):

The authors did a great effort to address almost all my comments, and significantly improved the paper. I believe that it is ready for publication.

Reviewer #3 (Remarks to the Author):

Brandt et al. use scRNA-Seq and single cell proteomics to study the effect of mutated KRAS and BRAF on colon organoids. Although both of these oncogenes are thought to work through activation of ERK they report that BRAFV600E induces a robust activation of ERK and the MAPK signaling pathway in all cells that express mutant BRAF. In contrast, mutant KRASG12V activates ERK only in a small subset of cells. The authors further suggest that this difference is mediated by DUSP proteins.

This is potentially an important finding that may have implications in our understanding of KRAS biology and in the development of KRAS inhibitors. However, at the current stage much more proof is needed. Specifically;

1. Ample research has demonstrated the importance of KRAS in cancer biology. Consistent with

the high frequency of KRAS mutations in human cancer in general and in colorectal cancer in particular KRAS has been demonstrated to induce highly robust cancer phenotypes as well as gene and protein changes in various models. In the current manuscript using both scRNA-Seq as well as single cell proteomics the authors show that induction of KRAS is similar to that of control cells. This is quite striking and if indeed could be confirmed would require re-examination of everything we know about KRAS. However, other more-simple explanations may shed light on these findings. For example, induction of KRAS in colonic organoids that have WT TP53 could induce oncogenic senescence. In other words, it is vital to demonstrate that over expression of KRAS is active in these cells.

2. In figure 1 the authors use patient derived organoids and show that activation of the MAPK pathway in these cells is graded and similar to that of KRAS WT organoids. In figure 2 they use transgenic mouse organoids and demonstrate that this is different in mutant BRAF. To make this point the authors should do the same CyTOF experiment in BRAF mutated patient derived organoids.

3. Based on proteomic analysis looking at differential expression in FRA-1 positive KRAS expressing cells the authors conclude that DUSP proteins may mediate this activation of KRAS. This is potentially an important finding, more experiments are needed to provide evidence for this hypothesis.

4. In figure 5A the authors sort FRA-1 positive cells as a marker for the population of colon organoids that show ERK activation in response to expression of mutated KRAS. This seems to be a very small population and based on the other results may represent background FRA-1 activity. The authors should show the same experiment in control (non-mutant KRAS expressing) cells.

Point-to-point response to the reviewers for Brandt et al. NCOMMS-18-17235R

The authors would like to thank the original reviewers for their positive comments on the revised manuscript. We also would like to thank the new reviewer for taking their time to assessing the revised study and raising important points. We have again, in this second round of major revisions, performed new experiments to address the new arguments.

Results of the new experiments have been added to the re-revised study as Figures S1b, c and S7. We have also recalculated the Modular Response Analysis in Fig. 7 to improve minor statistical aspects, which did not change the conclusions. We hope that we have addressed the new concerns in sufficient depth, and that all reviewers find the study now to be ready for publication.

Reviewers' comments:

Reviewer #1 (Remarks to the Author):

The authors have improved the manuscript in different sections; as suggested the authors have added data from patient-derived organoids which largely improves the manuscript. The clarification of the experimental systems used makes the manuscript easier to follow and puts the data in the context of other available scSeq datasets.

1. scSeq data from the patient-derived organoids complements the findings on mouse and improves the manuscript.

2. With regards to studying the KrasG12V mutant compared to G12D, though it is not the focus of the current study. I have to differ from the authors claim that these two mutants are similar. Though there hasn't been a head-to-head comparison in mouse models there is plenty of literature on characterizing the different Kras point mutations both biochemically and using isogenic cell lines. A few of these studies from the Westover and Prior Labs (Hunter et al., 2015; Hammond et al., 2015) are examples showing, that in-terms of biochemistry and proteome and phosphoproteome of these mutants is very different and that G12V has low activity than G12D and C. However, this does not make work from the authors on the G12V mutant irrelevant, it is important to acknowledge and be aware of work by others in understanding these mutations.

We thank the reviewer for the positive comments on our revised manuscript. We agree with the reviewer that different KRAS mutations can potentially have different downstream effects and discuss this now on p. 12. We also added the references suggested by the referee.

Minor comment:

There are a few typos during the rewrite: page 5, line 25, page 7, line 21

Thank you. We have corrected the typos.

Reviewer #2 (Remarks to the Author):

The authors did a great effort to address almost all my comments, and significantly improved the paper. I believe that it is ready for publication.

We thank the reviewer - we agree.

Reviewer #3 (Remarks to the Author):

Brandt et al. use scRNA-Seq and single cell proteomics to study the effect of mutated KRAS and BRAF on colon organoids. Although both of these oncogenes are thought to work through activation of ERK they report that BRAFV600E induces a robust activation of ERK and the MAPK signaling pathway in all cells that express mutant BRAF. In contrast, mutant KRASG12V activates ERK only in a small subset of cells. The authors further suggest that this difference is mediated by DUSP proteins.

This is potentially an important finding that may have implications in our understanding of KRAS biology and in the development of KRAS inhibitors. However, at the current stage much more proof is needed. Specifically;

1. Ample research has demonstrated the importance of KRAS in cancer biology. Consistent with the high frequency of KRAS mutations in human cancer in general and in colorectal cancer in particular KRAS has been demonstrated to induce highly robust cancer phenotypes as well as gene and protein changes in various models. In the current manuscript using both scRNA-Seq as well as single cell proteomics the authors show that induction of KRAS is similar to that of control cells. This is quite striking and if indeed could be confirmed would require re-examination of everything we know about KRAS. However, other more-simple explanations may shed light on these findings. For example, induction of KRAS in colonic organoids that have WT TP53 could induce oncogenic senescence. In other words, it is vital to demonstrate that over expression of KRAS is active in these cells.

We would like to break down this first point into three questions: 1) Is our finding of relatively small and/or cell type-specific effects of oncogenic KRAS in the intestinal epithelium in agreement with published literature? 2) Does oncogenic KRAS induce senescence in our experimental system and 3) is transgenic KRAS active in our experimental system?

In answer to question (1), we agree with the reviewer that KRAS is one of the most important (and most studied) oncogenes, as also shown by its high mutation frequency across many tissues, including the intestine. However, ample research has shown that KRAS is rather ineffective in initiating colorectal cancer (APC is the most common initiating mutation by a large margin)¹ or causing “robust cancer phenotypes” in humans, as brought up by the reviewer. For instance, the human colorectal lesion shown to the left has three different KRAS mutations and one NRAS mutation in four out of the six regions indicated, while two regions are free of RAS mutations (our unpublished data, in line with published studies^{2,3}). Even experienced pathologists cannot distinguish KRAS-mutant from KRAS-wildtype regions. In the mouse, oncogenic KRAS likewise does not immediately affect homeostasis in normal intestine or cause phenotypes in APC^{Min/-} adenoma, but leads to cancer phenotypes only with long latency^{4,5} (here, we study effects on signal transduction shortly after KRAS^{G12V} induction). Furthermore, there is disagreement in the literature to what extent transgenic oncogenic KRAS increases ERK phosphorylation in the mouse gut, as Sansom et al.⁴ found no upregulation of p-ERK by immunohistochemistry, while it was increased in Western blot in Janssen et al.⁵ and finally Haigis et al.⁶ found spatially restricted upregulation of p-ERK only at the top of colonic crypts after activation of KRAS^{G12D}. In agreement with the latter result, we find specific upregulation of p-ERK in late-stage enterocytes in transgenic organoids in our study (see Fig. 5d). Taken together, our findings do not require us to re-examine everything we know about KRAS in the intestine, but our data agree with the bulk of the scientific record on mutant KRAS in the normal and adenomatous intestinal epithelium in mouse and humans.

[REDACTED]

To provide an answer to question (2), we found that KRAS did not induced senescence-associated gene expression. We have added this analysis as Supplementary Figure 1b. Furthermore, we see that KRAS-induced cells were frequently Ki67-positive, again corroborating that KRAS^{G12V} does not induce senescence in the model system and time frame chosen for the experiments.

Red: KRAS^{G12V}-tdTomato; green: Ki67. Compound immunofluorescent images showing three z positions of an organoid, two days after KRAS transgene induction. White arrow heads indicate double-positive cells.

As the reviewer asked whether our KRAS construct is active (question 3), we now measured KRAS activity in a RAS loading assay. We find that activation of the KRAS^{G12V} transgene not only increased the total RAS pool of the cell, but also increased the active GTP-bound fraction to an even greater extent. We conclude that transgenic KRAS^{G12V} is active in our organoid system, in line with our finding of higher p-ERK levels in specific cell types. The data has been amended to the Results part of the manuscript, p. 6, and is shown as new Supplementary Figure S1c.

2. In figure 1 the authors use patient derived organoids and show that activation of the MAPK pathway in these cells is graded and similar to that of KRAS WT organoids. In figure 2 they use transgenic mouse organoids and demonstrate that this is different in mutant BRAF. To make this point the authors should do the same CyTOF experiment in BRAF mutated patient derived organoids.

As we have already stated in our previous point-to-point response concluding the first round of revisions, we unfortunately have no BRAF-mutant patient-derived organoid models available for direct comparison with the KRAS wildtype and mutant lines. That said, we agree with the reviewer that CyTOF-analysis of BRAF-mutant organoids would be an important addition to our paper. We have therefore now performed CyTOF analyses of BRAF-induced mouse organoids, similar to the previous experiments on KRAS-induced mouse organoids. In agreement with our other assays, we find that transgenic BRAF^{V600E} induced phosphorylation of ERK much stronger than KRAS^{G12V} and across all cell types. The new data is presented in Results p. 10, first paragraph and shown in the new Fig. S7 of the re-revised manuscript.

3. Based on proteomic analysis looking at differential expression in FRA-1 positive KRAS expressing cells the authors conclude that DUSP proteins may mediate this activation of KRAS. This is potentially an important finding, more experiments are needed to provide evidence for this hypothesis.

We agree that analysis of DUSP regulation and activity in the intestinal epithelium, in particular with regard to graded activities related to differentiation, would be an interesting and important topic for a further study. As a precedent of such a study, phosphatase networks interacting with ERK signaling have recently been found to underlie differentiation in the skin. This has been published in a beautiful eLife paper from Fiona Watts lab⁷, and we have cited this study in our revised discussion.

However, we firmly believe that such an endeavor is a complex study in its own right, and reaches far beyond the scope of this manuscript. We make it clear in the manuscript that the phosphatase network remains a hypothesis, see results (p.11, last paragraph): “As these phosphatases are known to dephosphorylate ERK ⁴⁶, we consider the DUSP1, DUSP5 and DUSP6 gene products as candidate mediators of attenuated MEK-ERK signal transmission that we observe in the differentiated cells of clusters 3-4.” And discussion (p.12, last paragraph): “A protein phosphatase network including DUSPs has recently been found to control ERK activity and differentiation in skin⁵¹. Our study suggests the existence of similar control mechanisms in the intestine.”

4. In figure 5A the authors sort FRA-1 positive cells as a marker for the population of colon organoids that show ERK activation in response to expression of mutated KRAS. This seems to be a very small population and based on the other results may represent background FRA-1 activity. The authors should show the same experiment in control (non-mutant KRAS expressing) cells.

The reviewer may have overlooked that Figure 5 already includes the requested control: these are labelled as grey cells in Fig. 5C-D. These cells were sorted from the induced control line, as suggested as a proper control by the reviewer.

The reviewer is right that in the KRAS^{G12V}-induced model we measure merged endogenous (or “background”, as termed by the reviewer) and KRAS^{G12V}-transgene-induced FIRE activity. We thus agree that the experiment shown in Fig. 5, by itself, would not be very revealing, as the endogenous and the KRAS^{G12V}-induced ERK/FIRE activities follow similar crypt-to-villus gradients. However, seen in context of the preceding and subsequent figures, the data presented in Figure 5 provides information on cell identity of the KRAS^{G12V}-induced cells having “extra” ERK/FIRE activity (Fig 4c) and being ERK phosphorylated in a cell-type-specific manner by KRAS^{G12V} (Fig. 6d).

References

1. Fearon, E. R. Molecular genetics of colorectal cancer. *Annu. Rev. Pathol.* **6**, 479–507 (2011).
2. Hershkovitz, D. *et al.* Adenoma and carcinoma components in colonic tumors show discordance for KRAS mutation. *Hum. Pathol.* (2014). doi:10.1016/j.humpath.2014.05.005
3. Gausachs, M. *et al.* Mutational heterogeneity in APC and KRAS arises at the crypt level and leads to polyclonality in early colorectal tumorigenesis. *Clin. Cancer Res.* (2017). doi:10.1158/1078-0432.CCR-17-0821
4. Sansom, O. J. *et al.* Loss of Apc allows phenotypic manifestation of the transforming properties of an endogenous K-ras oncogene in vivo. *Proc. Natl. Acad. Sci. U. S. A.* **103**, 14122–14127 (2006).
5. Janssen, K.-P. *et al.* APC and oncogenic KRAS are synergistic in enhancing Wnt signaling in intestinal tumor formation and progression. *Gastroenterology* **131**, 1096–1109 (2006).
6. Haigis, K. M. *et al.* Differential effects of oncogenic K-Ras and N-Ras on proliferation, differentiation and tumor progression in the colon. *Nat. Genet.* (2008). doi:10.1038/ng.115
7. Mishra, A. *et al.* A protein phosphatase network controls the temporal and spatial dynamics of differentiation commitment in human epidermis. *Elife* (2017). doi:10.7554/eLife.27356

REVIEWERS' COMMENTS:

Reviewer #3 (Remarks to the Author):

I believe that the authors answered most of the concerns raised and that this paper is ready for publication. Although, better experiment could be done to look at KRA induced oncogene senescence the other data presented here is sufficiently convincing.

Response to reviewer #3:

REVIEWERS' COMMENTS:

Reviewer #3 (Remarks to the Author):

I believe that the authors answered most of the concerns raised and that this paper is ready for publication. Although, better experiment could be done to look at KRA induced oncogene senescence the other data presented here is sufficiently convincing.

Thank you for taking your time to review our manuscript.